# LuminAIRe: Illumination-Aware Conditional Image Repainting for Lighting-Realistic Generation

Jiajun Tang[1,2]    Haofeng Zhong[1,2]    Shuchen Weng[1,2]    Boxin Shi[1,2*]

[1]National Key Laboratory for Multimedia Information Processing
[2]National Engineering Research Center of Visual Technology
School of Computer Science, Peking University
{jiajun.tang, hfzhong, shuchenweng, shiboxin}@pku.edu.cn

## Abstract

We present the il**Lumin**ation-**A**ware conditional **I**mage **Re**painting (LuminAIRe) task to address the unrealistic lighting effects in recent conditional image repainting (CIR) methods. The environment lighting and 3D geometry conditions are explicitly estimated from given background images and parsing masks using a parametric lighting representation and learning-based priors. These 3D conditions are then converted into illumination images through the proposed physically-based illumination rendering and illumination attention module. With the injection of illumination images, physically-correct lighting information is fed into the lighting-realistic generation process and repainted images with harmonized lighting effects in both foreground and background regions can be acquired, whose superiority over the results of state-of-the-art methods is confirmed through extensive experiments. For facilitating and validating the LuminAIRe task, a new dataset CAR-LUMINAIRE with lighting annotations and rich appearance variants is collected.

## 1  Introduction

Advanced image editing is in high demand across a multitude of applications, *e.g.*, old photo colorization [78, 32, 68], damaged image restoration [48, 73, 72], and artistic style transfer [22, 35, 70]. Recently, conditional image repainting (CIR) [67, 66, 58] has emerged as an innovative research topic, proven effective in controllable image editing while "freeing" users from the necessity of expert proficiency and retaining the "freedom" to actualize their creative visions for image modification. By utilizing provided attributes or textual descriptions, fine-grained strokes, and Gaussian noise to separately represent colors, contours, and texture conditions, users could insert generative objects with desired appearances in specified image positions, as shown in the blue line of Fig. 1.

Although CIR methods have made great progress in synthesizing photo-realistic and visually-pleasing conditional images by avoiding gradient vanishing pitfall [67], adopting flexible condition representation [66], and designing condition fusion modules [58], there is still a crucial element missing from the CIR task: making the synthesized results harmonized with the illumination of the scene, *e.g.*, spatially-varying dark and bright regions in accordance to the lighting condition in the background, physically-accurate highlight effects for highly-specular surfaces (shining objects), and perceptually-realistic shadow avoiding "floating objects" artifacts, as shown in the lower right example of Fig. 1.

Specifically, existing CIR methods handle image harmonization purely in 2D image space by estimating a pixel-wise color tone transformation of the repainted regions from the background regions. Current approaches use semantic parsing maps as "geometry" representations and do not exploit the lighting information contained in given background images, which prevents them from having

---

*Corresponding author.

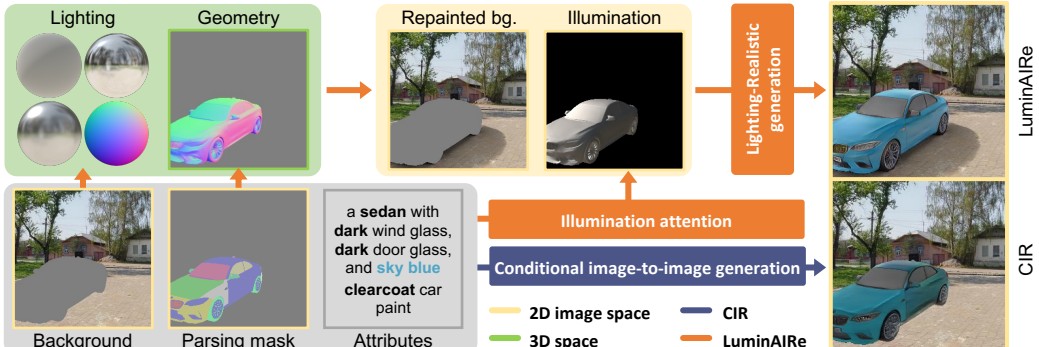

Figure 1: Illustration of proposed LuminAIRe task and result. Compared with the previous CIR task [58] (blue line) which takes all condition inputs[2] at once conducting a conditional image-to-image generation purely in 2D image space, LuminAIRe (orange line) exploits 3D lighting and geometry information and repaints both foreground (fg.) and background (bg.) regions via a lighting-realistic generation process. The 3D information is transformed back to 2D image space in the form of an illumination image, with the desired reflective properties obtained from an illumination attention module. LuminAIRe handles *(i)* surface shading, *(ii)* highlight effects, and *(iii)* realistic shadow in the repainted image (top right).

awareness of physically-based lighting in 3D space. To introduce physically-correct 3D lighting instead of hallucinated lighting effects into CIR results, there remain some major challenges: *(i)* The lighting condition in 3D space should be extracted from the limited field of view (limited-FoV) 2D LDR images; *(ii)* the lighting condition should be physically-correctly transformed back into 2D image space; *(iii)* a dataset suitable for learning-based solutions to the proposed task is needed.

To achieve *lighting-realistic generation* within the CIR pipeline in an illumination-aware manner, we hereby propose the task of il**Lumin**ation-**A**ware conditional **I**mage **Re**painting, denoted as **LuminAIRe**. We first lift geometry conditions from 2D parsing maps to 3D[1] normal maps using learning-based shape priors and estimate lighting conditions from limited-FoV LDR background images by designing a proper parametric representation. Then, we use physically-based reflection models to render *illumination candidate images* to capture possible lighting effects in 2D image space. With the help of *illumination attention module*, surface regions with different reflective properties are learned to adopt correct lighting effects in the resulting appearance. A dataset containing rich geometry and lighting annotations with abundant object variants is collected to facilitate the learning-based solution of the LuminAIRe task. As far as we know, we are the first to emphasize illumination-awareness in the image editing task of conditional image repainting.

Our contributions can be summarized as follows:

- introducing a new task of il**Lumin**ation-**A**ware conditional **I**mage **Re**painting (**LuminAIRe**) by exploiting the lighting information from background images;
- designing a full LuminAIRe pipeline that represents, extracts, converts, and injects lighting information to acquire more realistically repainted results; and
- collecting a new dataset CAR-LUMINAIRE with rich material and lighting condition variants for facilitating and validating the LuminAIRe task.

## 2 Related Work

Our method aims at introducing physical lighting constraints into generative image synthesis pipelines. In this section, we briefly review relevant works first and then discuss the relationships to our task.

**Controllable image synthesis.** Researchers have presented numerous works to synthesize images under the guidance of diverse user-provided conditions, *e.g.*, synthesizing specific object with category label [10, 43, 45, 75], transferring the texture from paintings to daily photos [22, 16, 35, 70], restoring

---

[1]Strictly speaking, the normal maps are in 2.5D. Here we use 3D to simply distinguish it from 2D.

[2]In this paper, attributes are shown in templated sentences for formatting, and texture is omitted for simplicity.

the colors of old photos [11, 12, 69, 68], and directly generating images from text descriptions [51, 50, 55, 56]. Recently, with the development of the condition injection mechanism [31, 47, 80, 34], researchers explore to control synthesized images with multiple cross-modality conditions, *e.g.*, condition guided image inpainting [44, 55], controllable person synthesis [54, 65], and inversion-based style transfer [79]. However, few works focus on synthesizing images strictly following lighting conditions. Following DIH-GAN [6] that considers introducing illumination estimation in harmonization task that adjusts the highlight of the inserted given object, we further explore the lighting condition in synthesizing illumination-consistent objects under the guidance of multiple cross-modality conditions.

**Conditional image repainting and image harmonization.** Conditional image repainting (CIR) aims at synthesizing reasonable visual content on an existing image, where the generated visual content should both meet the requirement of the user-provided conditions (*e.g.*, color, geometry, and texture) and in harmonization with the existing background image. The first CIR task is proposed in MISC [67] for person image synthesis, where the foreground person image is synthesized first and then composited with the background. Weng *et al*. [66] design the semantic-bridge attention mechanism which allows more freely expressed color conditions by the users in text. UniCoRN [58] breaks the two-stage dependency and proposes a unified architecture that achieves more visually pleasing results. Despite recent achievements made by previous works in condition consistency, existing CIR models suffer from the issue of illumination inconsistency: although techniques such as color tone transform are applied, the lighting from the given background and on the generated visual contents often differ a lot, making lighting effects in the image rather unrealistic, such as incorrect shading, highlights, and shadows. In this paper, we address this issue by exploiting lighting and shape constraints in 3D space, which allows a more physically-correct rendering processing for generating lighting effects. Image harmonization methods [23, 24, 25, 14, 42, 62, 59], with a similar goal of CIR to realistically composite image foreground and background regions, have focused on illumination harmonization recently [6, 8]. However, this thread of works has poor control of visual content in foreground regions and may fail to preserve the color tone in background regions as they were.

**Lighting representation and estimation.** Achieving illumination-aware synthesis/repainting requires appropriate lighting representation and estimation from images. Lalonde *et al*. [37] is the first to use shadows, shading, and sky appearance variations observed in the image to infer outdoor lighting. A physics-based Hošek-Wilkie (HW) sky model [29, 30] is proposed to recover HDR parameters for deep outdoor lighting estimation [28]. A more weather-robust Lalonde-Matthews (LM) model [38, 77] is then proposed to cover more comprehensive lighting conditions in the outdoor environment. More recently, a learning-based lighting representation [27] is used on a large sky panorama dataset [36] with an autoencoder network. The encoder-decoder framework is further proposed [39] to estimates lighting as a spherical HDR lighting map. HDSky [74] and SOLD-Net [60] disentangle several physically meaningful attributes into separate learned latent spaces by hierarchical autoencoders and make the estimation editable. Parametric models such as spherical harmonic (SH) coefficients [7, 21] and spherical Gaussian (SG) [19, 40] are also widely used, especially in indoor scenes. Gardner *et al*. [20], NeurIllum [57], and SOLID-Net [81] design sophisticated networks to hallucinate the missing parts in the panoramic view and predict lighting as environment maps. 3D volumetric lighting representations are also widely used in recent works, which facilitate the lighting-realistic scene editing for indoor [41] and outdoor [64] scenes, however heavily require computation resources. Considering the demand for lighting-realistic generation, we propose a parametric lighting representation for outdoor scenes that is both easy to predict and simple to use.

## 3 Problem Formulation

For self-containedness, we briefly review the CIR formulation before introducing ours.

### 3.1 Preliminaries about CIR

The previous CIR tasks [66, 67, 58] aim at generating the repainted image $y^{\mathrm{r}}$ by repainting certain regions in an image $x \in \mathbb{R}^{3 \times H \times W}$ according to user-specified conditions in different modalities: $x^{\mathrm{g}}$, $x^{\mathrm{p}}$, $x^{\mathrm{c}}$, and $x^{\mathrm{b}}$ for the "geometry", "texture", "color", and background conditions respectively.

In their works, the "geometry" condition $x^{\mathrm{g}} \in \mathbb{L}^{N_{\mathrm{g}} \times H \times W}$ is a binary semantic parsing mask, where $N_{\mathrm{g}}$ is the number of possible parts of the visual content to be repainted and $\mathbb{L} = \{0, 1\}$; the "texture"

condition $x^p \sim \mathcal{N}(0,1)$ is a Gaussian noise; the "color" condition can be represented as attributes $x^c \in \mathbb{L}^{N_c \times N_v}$ or text descriptions $x^c = \{x_t^c\}_{t=1}^{N_L}$, where $N_c$, $N_v$, and $N_L$ represent the numbers of attributes and available choices, and the length of the user-inputted sentences, respectively; the background condition $x^b \in \mathbb{R}^{3 \times H \times W}$ is the image of background region with respect to the repainted region as foreground region, $i.e.$, $x^b = (1-m) \odot x$, where the binary mask $m$ indicating foreground region can be directly acquired from the parsing mask $x^g$, as shown in lower left of Fig. 1.

The repainted image $y^r$ can be further decomposed as a blending of repainted foreground image $y^f$ and repainted background image $y^b$:

$$y^r = m \odot y^f + (1-m) \odot y^b. \tag{1}$$

Previous works assume unchanged background region, $i.e.$, $y^b = x^b$, leaving the key question of CIR tasks as generating realistic foreground region $y^f$ constrained by given conditions:

$$y^f = F^G(x^g, x^p, x^c, x^b), \tag{2}$$

where previous works ignore clues in 3D space and implement the generation pipeline $F^G$ as a *conditional image-to-image generation* purely in 2D image space. To make the repainted image harmonized as a whole, previous works [58, 67] design additional harmonization modules to adjust the color tone of intermediate repainting result based on clues in $x^b$.

## 3.2 Formulation of LuminAIRe

However, the image-based harmonization modules have limited representation ability for complex lighting effects ($e.g.$, varying shading and shiny surfaces) due to a lack of 3D representation. Besides, directly using $x^b$ as $y^b$ in Eq. (1) may neglect possible light transport effects ($e.g.$, shadows) introduced by the repainted region as its corresponding behaviors in the 3D real world might be.

As illustrated by the rendering equation [33], a physically-correct and -realistic appearance of an object is derived from its *geometry*, *reflective property*, and omnidirectional *environment lighting* in 3D space. Therefore, to make the repainted image $y^r$ more *lighting-realistic*, the repainted foreground $y^f$ should also be conditioned by the lighting condition $L$ and geometry condition $G$ in 3D space:

$$y^f = F^F(x^g, x^p, x^c, x^b, L, G). \tag{3}$$

Given $L$ and $G$ in 3D space, a proper 2D representation $x^i$ containing both the information from $L$ and $G$ should be derived for compatibility with current image generation architectures:

$$x^i = R^i(L, G), \tag{4}$$

and then the *lighting-realistic generation* for foreground $y^f$ can be rewritten as:

$$y^f = F^F(x^g, x^p, x^c, x^b, x^i). \tag{5}$$

Similarly, the repainted background $y^b$ should also be conditioned on $x^i$ to recover lighting effects:

$$y^b = F^B(x^b, x^i). \tag{6}$$

The limited-FoV background image $x^b$ itself is a partial observation of environment lighting and thus can provide clues about $L$. Therefore, the lighting condition $L$ can be inferred in the form of:

$$L = F^L(x^b). \tag{7}$$

Similarly, by finding the shape priors of certain types of objects, the 3D geometry condition $G$ can be lifted from its "2D flattened version", $i.e.$, parsing mask $x^g$:

$$G = F^{Geo}(x^g). \tag{8}$$

Moreover, in our LuminAIRe formulation, we extend the attributes $x^c$ beyond colors, which allows the users to describe the *reflective property* and have control over the lighting effects of repainting results. A sample of attributes is shown as the **bold text** in the lower left of Fig. 1.

As aforementioned, both the repainted foreground $y^f$ and background $y^b$ are given by the *lighting-realistic generation* in our LuminAIRe formulation, which leads to more realistic and harmonized results than traditional CIR pipelines [58], as shown in Fig. 1.

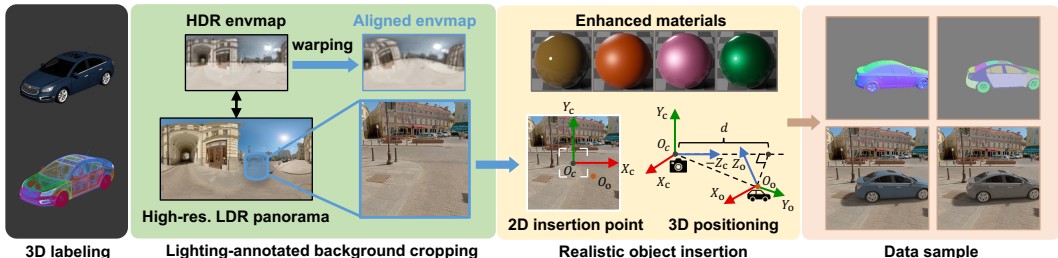

Figure 2: Data preparation process of the CAR-LUMINAIRE dataset.

# 4 Data Preparation

To tackle the data shortage issue, we create the first dataset suitable for the LuminAIRe task, named CAR-LUMINAIRE, with its data preparation process and data sample shown in Fig. 2.

**3D car models with hierarchical semantic labeling.** Collecting large-scale real data for learning-based LuminAIRe methods is infeasible since the geometry and lighting capture in 3D space requires specialized equipment and extensive human labor. Therefore, here we resort to computer graphics techniques to create photo-realistic synthetic data. The cars are chosen as the foreground objects for the obviousness of lighting effects and the availability of high-quality synthetic models. We collect 198 detailed 3D car models in 17 different categories from online model stores [2, 4] and then label the parts of the models in 3D space, which allows us to get the accurate parsing mask in 2D image space from any viewpoint. Following the common structure of vehicles, we divide the car models into 35 semantic part labels. The part labels are organized in a hierarchical way (*e.g.*, the *door window* is a sub-part of the *door*) to accommodate car models in different granularity. Besides 3D labeling, we manually adjust the scales of each model to fit the real-world dimensions.

**Background images with lighting annotations.** Then we prepare background images with known lighting annotations. Here we use the SUN360-HDR dataset [27, 76], which contains HDR panoramic environment maps (envmaps) corresponding to the LDR panoramas of outdoor scenes in the SUN360 dataset [71]. Limited field-of-view (limited-FoV) background images are cropped from the LDR panoramas with virtual cameras of randomized FoVs and camera poses. For each cropped background image, the corresponding HDR envmap in the SUN360-HDR dataset [27, 76] is warped to align with the viewing direction of the virtual camera. Background images unsuitable for realistic object insertion are manually filtered out, leaving 1,321 images of diverse scenes and lighting conditions.

**Enhanced data rendering with realistic placement.** For each background image, we randomly select insertion points within the central region of the "placeable flat ground" marked by an off-the-shelf segmentation toolbox [15]. Then, for each 2D insertion point in the image, we calculate the relative transformation from the camera coordination $O_c$ to the local coordination of the object $O_o$ from the depth $d$ and the normal $Z_o$ estimated by depth [52, 53] and normal [5] estimation methods. With the aligned envmaps and the ray-tracing based Blender [3] Cycles rendering engine, physically-correct lighting effects can be rendered into the composited images. In the rendering process, besides the original materials of the models, several physics-based rendering (PBR) car paint materials are randomly applied for more appearance variants, especially in lighting effects; besides, the inserted models are randomly rotated around $Z_o$ axis for more geometry variants. The rendered images are filtered to ensure reasonable pixel portions of both foreground and background regions. At last, 52,581 composited images at the resolution of $256 \times 256$ are collected, accompanied by parsing mask and normal map annotations, as shown in the data sample of Figure 2.

# 5 Method

To realize the LuminAIRe formulation, we first estimate 3D lighting and geometry from background images and parsing masks (Sec. 5.1). Then the lighting information is injected into the lighting-realistic generation process as illumination images (Sec. 5.2). By further introducing hierarchical labeling enhancement (Sec. 5.3), our method can generate reasonable results even with coarse-level parsing masks. Our pipeline is shown in Fig. 3, with detailed network architectures and loss functions for network modules in supplementary materials.

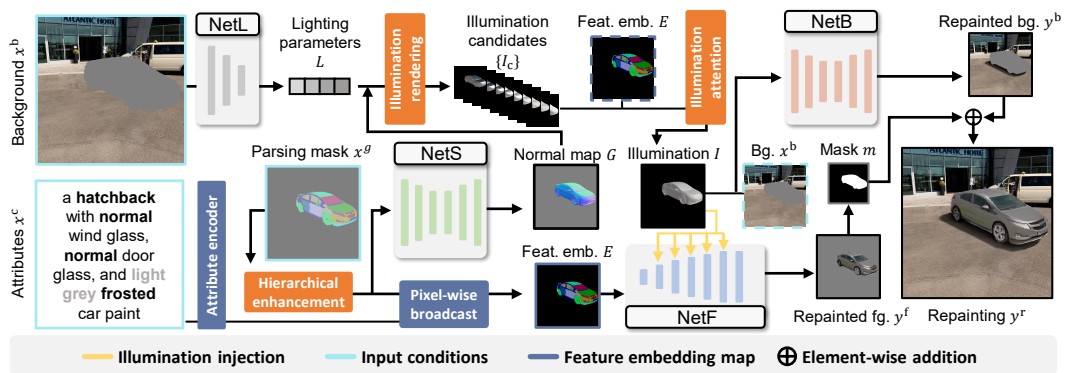

Figure 3: Overview of LuminAIRe pipeline.

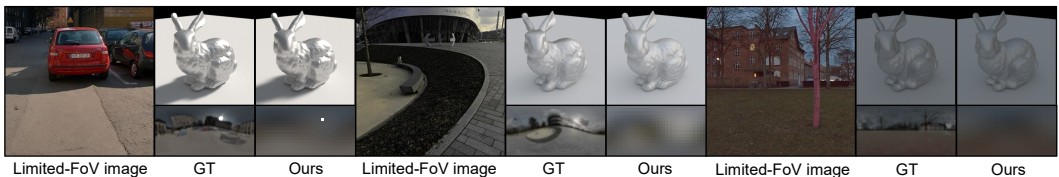

Figure 4: Our lighting representation can capture most of the lighting effects in all weather conditions.

## 5.1 Estimating 3D Information with Learning-based Priors

Our CAR-LUMINAIRE dataset consists of outdoor scene images, where the lighting can be approximately decomposed into the high-frequency sunlight and the low-frequency ambient light [60]. Accordingly, we model the lighting condition $L$ as the addition of a directional light and a 2-nd order spherical harmonics (SH) lighting, which can be specifically described as lighting parameters:

$$L = \{z_{\text{vis}}, z_{\text{int}}, z_{\text{ang}}, c_{\text{sun}}, l_{\text{sun}}, \sigma_{\text{SH}}\}, \tag{9}$$

where $z_{\text{vis}} \in \{0, 1\}$ is the sun visibility, $z_{\text{int}}$ is the intensity of sunlight, $z_{\text{ang}}$ describes the "size" of the sun (in solid angle formally), $c_{\text{sun}} \in \mathbb{R}^3$ is the normalized sun color in RGB channels, $l_{\text{sun}} \in \mathbb{R}^2$ indicates the sun position, and $\sigma_{\text{SH}} \in \mathbb{R}^{3 \times 9}$ is the 2-nd order SH coefficients for RGB channels.

As shown in Fig. 4, the parametric representation[3] in Eq. (9) can well fit real-world lighting in sunny, cloudy, and low light conditions. On the other hand, the proposed parametric lighting representation is convenient for network prediction. Here we design a *NetL* to serve as $\text{F}^{\text{L}}$ in Eq. (7), where $l_{\text{sun}}$ is estimated by a classification task and other parameters are estimated by regression tasks. To apply our method to other types of background scenes, specifically tailored lighting representations can be directly adopted, without modification to our underlying formulation of LuminAIRe.

For 3D geometry, we use the normal map $G \in \mathbb{R}^{3 \times H \times W}$ as the representation where each pixel indicates the surface normal direction $\boldsymbol{n}$ at that surface point in 3D space. For certain types of objects, there exist strong shape priors (such as sedans and hatchbacks), which can be learned in a supervised way. Similarly, a *NetS* of encoder-decoder structure is further proposed to serve as $\text{F}^{\text{Geo}}$ in Eq. (8).

## 5.2 Injecting Lighting Information using Illumination Images

To bridge the 3D lighting and geometry with 2D images, the rendering equation [33] is a handy tool to serve as $\text{R}^{\text{i}}$ in Eq. (4), which physically models the image formation process as the light reflection:

$$L_{\text{o}}(\boldsymbol{\omega}_{\text{r}}) = \int_{\Omega_{\boldsymbol{n}}} L_{\text{i}}(\boldsymbol{\omega}_{\text{i}}) f_{\text{r}}(\boldsymbol{\omega}_{\text{i}}, \boldsymbol{\omega}_{\text{r}})(\boldsymbol{n} \cdot \boldsymbol{\omega}_{\text{i}}) \mathrm{d}\boldsymbol{\omega}_{\text{i}}, \tag{10}$$

where $L_{\text{i}}(\boldsymbol{\omega}_{\text{i}})$ is the environment lighting from direction $\boldsymbol{\omega}_{\text{i}}$, $L_{\text{o}}(\boldsymbol{\omega}_{\text{r}})$ is the reflected lighting toward direction $\boldsymbol{\omega}_{\text{r}}$, $\Omega_{\boldsymbol{n}}$ is the visible hemisphere determined by surface normal $\boldsymbol{n}$, and $f_{\text{r}}(\boldsymbol{\omega}_{\text{i}}, \boldsymbol{\omega}_{\text{r}})$ describes the reflective properties of all possible combination of incoming and outgoing directions.

---

[3]Lighting parameters are converted back to tone-mapped HDR environment maps for visualization.

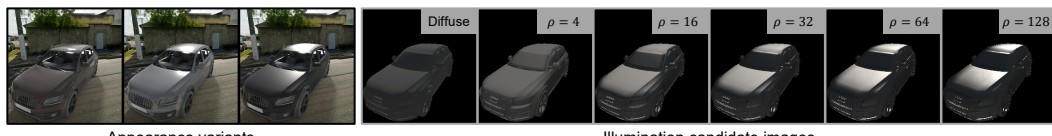

Appearance variants       Illumination candidate images

Figure 5: Our illumination candidate images can cover realistic lighting effects in appearance variants.

For a certain image pixel with the known camera viewing direction $\boldsymbol{v}$, ideally, the pixel intensity can be calculated as $L_\mathrm{o}(-\boldsymbol{v})$, and accurate lighting effects can be calculated as *illumination images*.

However, with $L$ and $G$ estimated from input conditions, $f_\mathrm{r}(\boldsymbol{\omega}_\mathrm{i}, \boldsymbol{\omega}_\mathrm{r})$ still remains unknown. Therefore, in a similar spirit to Gao *et al.* [18] and Pandey *et al.* [46], instead of directly calculating the actual illumination image, we use a set of uncolored "standard materials" as $f_\mathrm{r}$ in Eq. (10) and render corresponding *illumination candidate images* $\{I_\mathrm{c}\}$. For the physics-based rendering of $\{I_\mathrm{c}\}$, we use the Lambertian refectance model $f_\mathrm{diff}(\boldsymbol{\omega}_\mathrm{i}, \boldsymbol{\omega}_\mathrm{r}) = 1/\pi$ and normalized Blinn-Phong model [9] $f_\mathrm{spec}(\boldsymbol{\omega}_\mathrm{i}, \boldsymbol{\omega}_\mathrm{r}) = (\rho+4)(\boldsymbol{n}\cdot\boldsymbol{h})^\rho/8\pi$ with $M$ different values of roughness $\rho$, where $\boldsymbol{h} = \boldsymbol{\omega}_\mathrm{i}+\boldsymbol{\omega}_\mathrm{r}/||\boldsymbol{\omega}_\mathrm{i}+\boldsymbol{\omega}_\mathrm{r}||$ is the half vector. At last, we have $\{I_\mathrm{c}\} = \{I_\mathrm{diff}\} \cup \{I_\mathrm{spec}^{\rho_i}\}_{i=1}^{M}$.

As shown in Fig. 5, most lighting effects in different appearance variants can be covered by the linear combinations of the pre-computed $\{I_\mathrm{c}\}$. However, it's worth noting that the correspondence of the appearance image and $\{I_\mathrm{c}\}$ may vary pixel-wisely (*e.g.*, the tires, hood, and windshield have different reflective properties thus different lighting effects). Accordingly, we design an *illumination attention* module $\mathrm{A}^\mathrm{I}$ to estimate the combination coefficient maps $C_\mathrm{I} = \mathrm{A}^\mathrm{I}(E)$ for each image pixel, where $E$ is the feature embedding map containing information of both part labels and part-associated attributes in a pixel-aligned way. After the illumination image $I$ derived as $I = \sum_{i=1}^{M+1} C_\mathrm{I}^i \odot I_\mathrm{c}^i$, which covers lighting effects of parts with different materials , we use $I$ as $x^\mathrm{i}$ in Eq. (5) and conduct lighting-realistic generations of foreground and background regions using our proposed *NetF* and *NetB* respectively. For *NetF*, we adopt the network backbone of $\mathrm{F}^\mathrm{G}$ in UniCoRN [58], and the illumination image $I$ is injected in a similar way as other conditions in 2D image space at different resolutions. The *NetB* is also an encoder-decoder architecture, serving as $\mathrm{F}^\mathrm{B}$ in Eq. (6). We adopt the same loss functions for *NetF* as used in UniCoRN [58].

### 5.3 Generating Realistic Results from Coarse Parsing Masks

As mentioned in Sec. 4, the parsing masks in our Car-Luminaire dataset can be very coarse, which also reflects the possible application scenarios when the user only specifies interested parts. Previous CIR formulations may fail to generate realistic results in regions without fine-grained labels since their generation follows a strictly pixel-wise semantic mapping between labels and images. We hereby introduce a *hierarchical labeling enhancement*: randomly coarsening the input parsing mask at training time (*e.g.*, *door glass* label becomes *door* label) and encouraging the fine-grained parts (*door glass*) to be generated. Besides, the part-associated attributes of lower-level parts (*door glass*) should be also associated with their upper-level parts (*door*) to avoid loss of condition in attributes $x^\mathrm{c}$, which can be done by modifying the association matrix [58] $A \in \mathbb{L}^{N_\mathrm{c} \times N_\mathrm{g}}$ accordingly.

## 6 Experiments

In this section, we conduct comparisons with state-of-the-art methods and validate our design with an ablation study and a robustness test. Please see supplementary materials for implementation details.

### 6.1 Comparison with State-of-the-art Methods

**Baseline methods.** We conduct quantitative and qualitative comparisons with three state-of-the-art CIR methods (**UniCoRN** [58], **Weng *et al.*** [66], and **MISC** [67]) and a most-relevant conditional image generation method (**Pavllo *et al.*** [49]). Among them, modifications are made for **Pavllo *et al.*** [49] and **Weng *et al.*** [66] to accept conditions represented as attributes.

**Quantitative metrics.** Following previous work [58], we adopt Fréchet inception distance (FID) [26] for assessment of perception quality, R-precision [66] for assessment of alignment between generated

images $y^r$ and given attributes $x^c$, and M-score [61] for assessment of authenticity. We use the latest manipulation detection model [17, 13] for calculating the M-score [61]. We also report the structural similarity index (SSIM) [63] for comparing the major image structure with the reference image.

Table 1: Comparison with the state-of-the-art methods and variants of our proposed method. Quantitative evaluation scores and user study results are shown. ↑ (↓) means higher (lower) is better. "Real." and "Har." are abbreviations of "Realistic" and "Harmonized".

| Method | Quantitative Evaluation | | | | User Study | |
| | FID ↓ | R-prcn ↑ | M-score ↓ | SSIM ↑ | Real. ↑ | Har. ↑ |
|---|---|---|---|---|---|---|
| MISC [67] | 53.84 | 34.94% | 31.23 | 0.6660 | 0.25% | 0.28% |
| Weng *et al*. [66] | 38.12 | 46.66% | 30.84 | 0.6697 | 0.85% | 0.85% |
| Pavllo *et al*. [49] | 9.29 | 56.98% | 36.77 | 0.7050 | 43.00% | 36.72% |
| UniCoRN [58] | 11.55 | 62.13% | 29.72 | 0.6940 | 7.78% | 9.90% |
| LuminAIRe (Ours) | **4.62** | **74.13%** | **13.68** | **0.7211** | **48.12%** | **52.25%** |
| Ours-H | 5.83 | 63.27% | 13.97 | 0.7163 | — | — |
| Ours-HA | 6.31 | 63.94% | 13.95 | **0.7214** | — | — |
| Ours-HAI | 8.00 | 62.13% | 15.83 | 0.7054 | — | — |

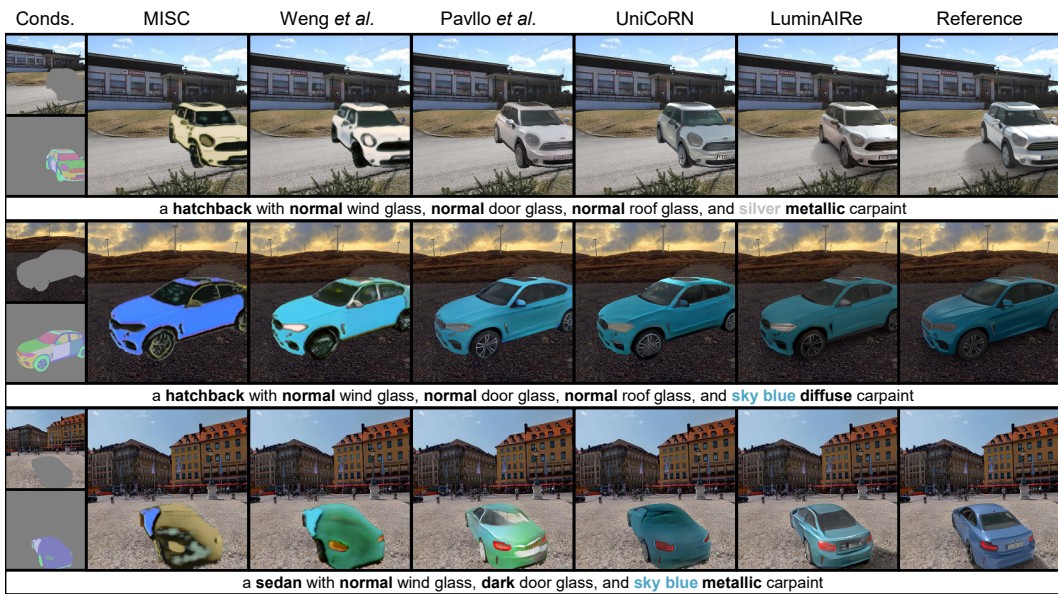

Figure 6: Qualitative comparison with the state-of-the-art methods, with given conditions (conds.).

The scores in Tab. 1 and the second and the third columns of Fig. 6 show that results of **MISC** [67] and **Weng *et al*.** [66] are far from lighting-realistic with "crayon-drawing-like" appearances, since the color tone transform is not applied [66], or conducted in a two-phase manner [67]. As shown in the fourth column of Fig. 6, **Pavllo *et al*.** [49] tend to generate foreground regions in flat shadings with fewer texture patterns, which makes its results generally look reasonable when only focusing on foreground regions or in low light or cloudy scenes (as indicated by the FID and user study results), but computer vision models can easily find the disharmony due the sharp boundaries between foreground and background regions [58], as also indicated by the worst M-score. **UniCoRN** [58] fails to generate correct lighting effects from its unified color tone transform (the first row), therefore tends to hallucinate highlights at the top of cars regardless of lighting in background regions (the second row). The hallucinated lighting effects along with the undesired texture pattern on car bodies drastically damage the perceptual preferences, as confirmed by the FID score and user study results in Tab. 1. **LuminAIRe** generates realistic lighting effects close to the reference images in both sunny (the first and the third rows) and cloudy (the second row) scenes of specified materials and even when a coarse-level parsing mask is given (the third row), with a large margin in all quantitative metrics compared with baseline methods. **LuminAIRe** also learns to avoid the undesired texture pattern with the hints of the smoothly varied shading in the illumination images (Fig. 7).

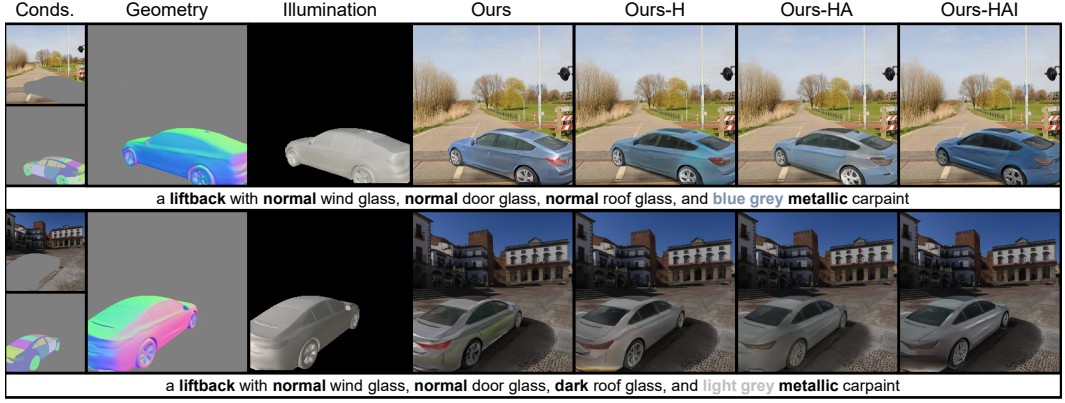

Figure 7: Ablation study for three variants of our proposed method, with given conditions (conds.).

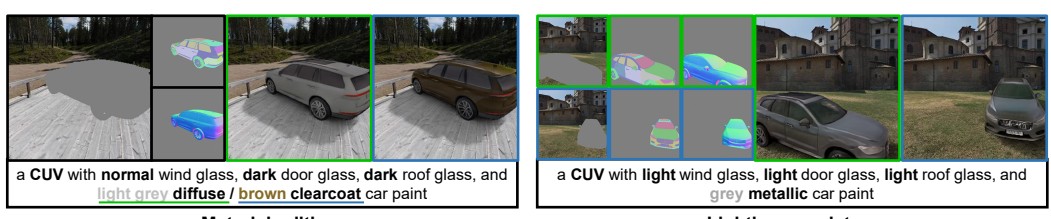

Figure 8: Our method can generate realistic lighting effects with given materials (left), which are consistent across different geometry conditions (right). Green and blue boxes mark individual cases.

## 6.2 Evaluations

**Ablation study.** We conduct an ablation study with three variants of **Ours**: *(i)* **Ours-H**, *(ii)* **Ours-HA**, and *(iii)* **Ours-HAI**, where "-H", "-A", and "-I" mean disabling the hierarchical labeling enhancement, the illumination attention, and the illumination injection for the foreground, respectively.

The hierarchical labeling enhancement is confirmed helpful in generating realistic results with coarse-level parsing masks, as shown in Fig. 7 and the third row of Fig. 6, where **Ours** generates more consistent and better repaintings at regions with no specified part labels (marked in blue purple), which is also demonstrated by the FID and R-prcn score in Tab. 1. The second row of Fig. 7 shows an example where the lack of illumination attention module wrongly renders a diffuse appearance, with further evidence from the drop of FID from **Ours-H** to **Ours-HA** in Tab. 1. It's quite obvious from Tab. 1 and Fig. 7 that the illumination injection helps foreground generation by comparing **Ours-HA** and **Ours-HAI**. From **UniCoRN** to **Ours-HAI**, the improvements in FID score and M-score validate the contribution of the lighting-realistically generated background.

Besides, **Ours-HA** gets an unexpectedly good SSIM score. It's possibly because a slight misalignment of lighting effects (especially highlights) due to errors in lighting or geometry estimation would lead to a considerable drop in the SSIM score (which honestly measures the pixel-wise difference) but with very little harm to the lighting-realistic perception (as indicated by the FID and M-score).

**User study.** We also conduct a user study with 20 volunteers on the Amazon Mechanical Turk [1] platform, where 200 sets of results randomly drawn from the test set are shown and volunteers are asked to choose one in each set with *(i)* the most *realistic* foreground and *(ii)* the most *harmonized* lighting. The results of the user study in Tab. 1 are basically aligned with the trending of FID and SSIM scores in quantitative evaluation, showing that repainting results of our **LuminAIRe** are most favored subjectively, with a greater lead in realistic and harmonized lighting perception.

**Robustness Test.** Fig. 8 shows the robustness of our method to varying materials and geometry conditions, where different materials and geometry conditions are correctly handled with realistic lighting effects accordingly and consistently generated. To test the robustness of our method to varying parsing masks (*e.g.*, casually-drawn parsing masks), we compare in Fig. 9 the repainting

results of from the input parsing masks before and after the disturbing, where the boarders are randomly extended and the inner structures are coarsened. To test the robustness of our method to varying lighting conditions, we conduct an experiment where the estimated lighting conditions are rotated clockwise while all other conditions are left unchanged. The results in Fig. 10 show that our method correctly handles most of the lighting rotations in the sense of the lighting effects on the foreground objects and the shadow effects in the background regions. The repainting results in the second column with no lighting conditions given ("No light") further validate the effectiveness of our illumination injection module. To test the robustness of our method to varying background conditions, we also show the results of in-the-wild examples in the supplemental material.

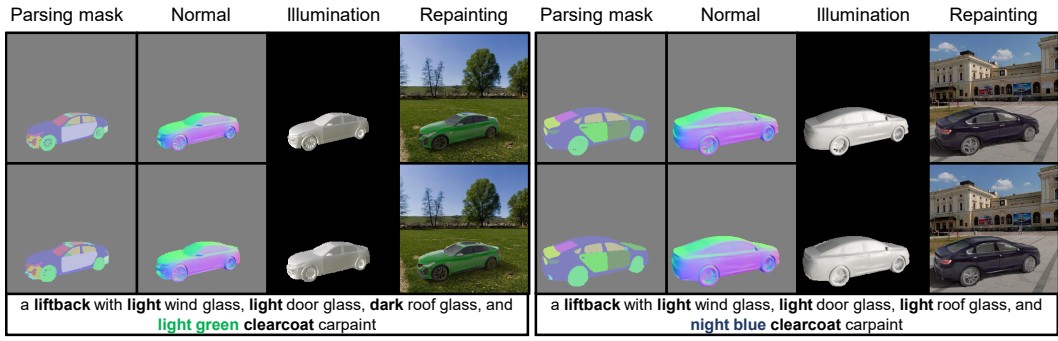

Figure 9: Qualitative results of normal maps, illumination images and repaintings using original (first row) and disturbed (second row) parsing masks as input conditions. Backgrounds are omitted here.

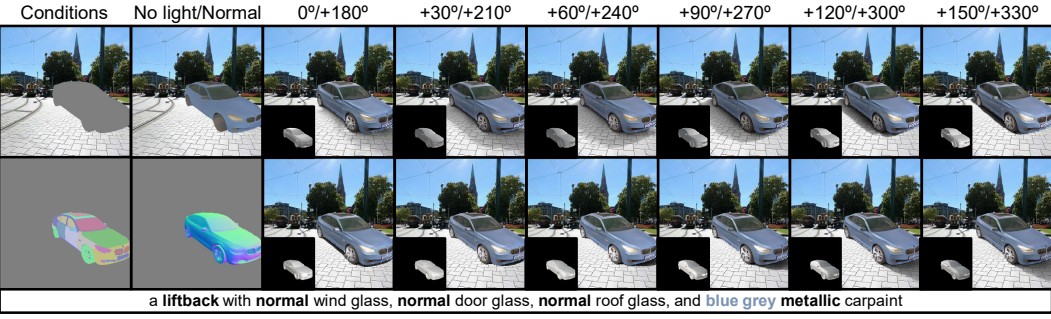

Figure 10: Qualitative results of repaintings and illumination images as the estimated lighting rotates.

## 7 Conclusion

In this paper, we introduce the task of LuminAIRe for the realistic generation of lighting effects. The synthetic CAR-LUMINAIRE dataset is collected for the newly proposed task. Extensive experiments and the user study confirm that our method achieves perceptually more lighting-realistic and harmonized repainting results compared with the state-of-the-art methods. The effectiveness and consistency of our illumination-aware design are shown in the robustness test.

**Limitations and future works.** In this paper, only the results of cars as foreground objects are shown, resulting from the inadequate feasibility of data collection. Besides, our model can not handle complex thin structures and some translucent glass materials very well, which are not well covered by our synthetic data for now. As a single-image-based method for generic outdoor scenes, our method currently ignores the non-local inter-reflections with other objects and focuses on the shadows cast directly on the ground. Therefore, datasets of richer object categories and finer details will be helpful to boost the training of learning-based methods. Combining the lighting constraints with the newly emerged latent diffusion models [55] would also be an interesting direction for our future work.

**Acknowledgement.** This work is supported by the National Natural Science Foundation of China under Grant No. 62136001, 62088102.

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
