# LuminAIRe: Illumination-Aware Conditional Image Repainting for Lighting-Realistic Generation (Supplemental Material)

**Jiajun Tang**[1,2]    **Haofeng Zhong**[1,2]    **Shuchen Weng**[1,2]    **Boxin Shi**[1,2*]

[1]National Key Laboratory for Multimedia Information Processing
[2]National Engineering Research Center of Visual Technology
School of Computer Science, Peking University
{jiajun.tang, hfzhong, shuchenweng, shiboxin}@pku.edu.cn

In this supplementary material, we provide more information about our data collection, implementation details, and network architectures. We also show additional results on our CAR-LUMINAIRE dataset and in-the-wild data.

## 8    Appendix

### 8.1    Details on the CAR-LUMINAIRE Dataset

When calculating the object's local coordinate system $O_o$, we assume that $X_c \parallel X_o$, that is to say, we assume that the ground is approximately horizontally level, which is satisfied at most times.

We divide the car into 35 classes of parts, of which the hierarchy and color coding are shown in Fig. 11.

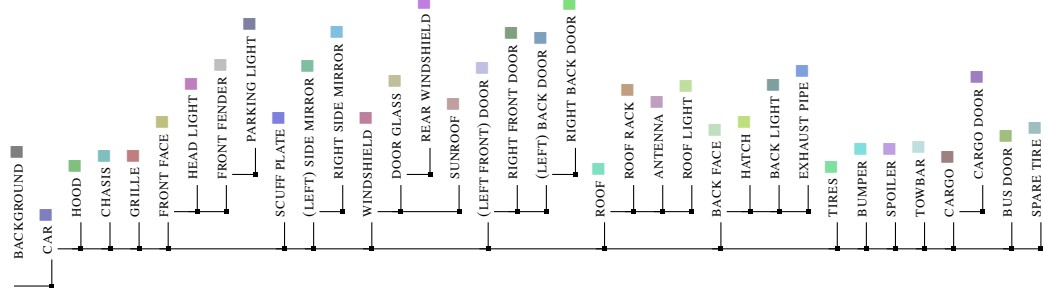

Figure 11: The hierarchy and color coding of the part labels used in the CAR-LUMINAIRE dataset.

The attribute of car models is also manually annotated when labeling the parts of models. We mark $N_c = 6$ major part-related attributes (car type, wind glass darkness, door glass darkness, roof glass darkness, car paint color, and car paint type) with $N_v = 81$ available choices. The $N_g$ is set as 35 following our hierarchical labeling. The relationship between parts and corresponding attributes is represented as an association matrix [14] $A \in \mathbb{L}^{N_c \times N_g}$.

We use a randomly chosen camera pitch in $[-15°, 15°]$ and FoV in $[25°, 66°]$ for the background image cropping. The same FoV is used in image rendering for view consistency. For each combination of the background image and geometry condition, we render one image of the original car model and two images of variants with randomly chosen enhanced car paint materials. When splitting the

---

*Corresponding author.

37th Conference on Neural Information Processing Systems (NeurIPS 2023).

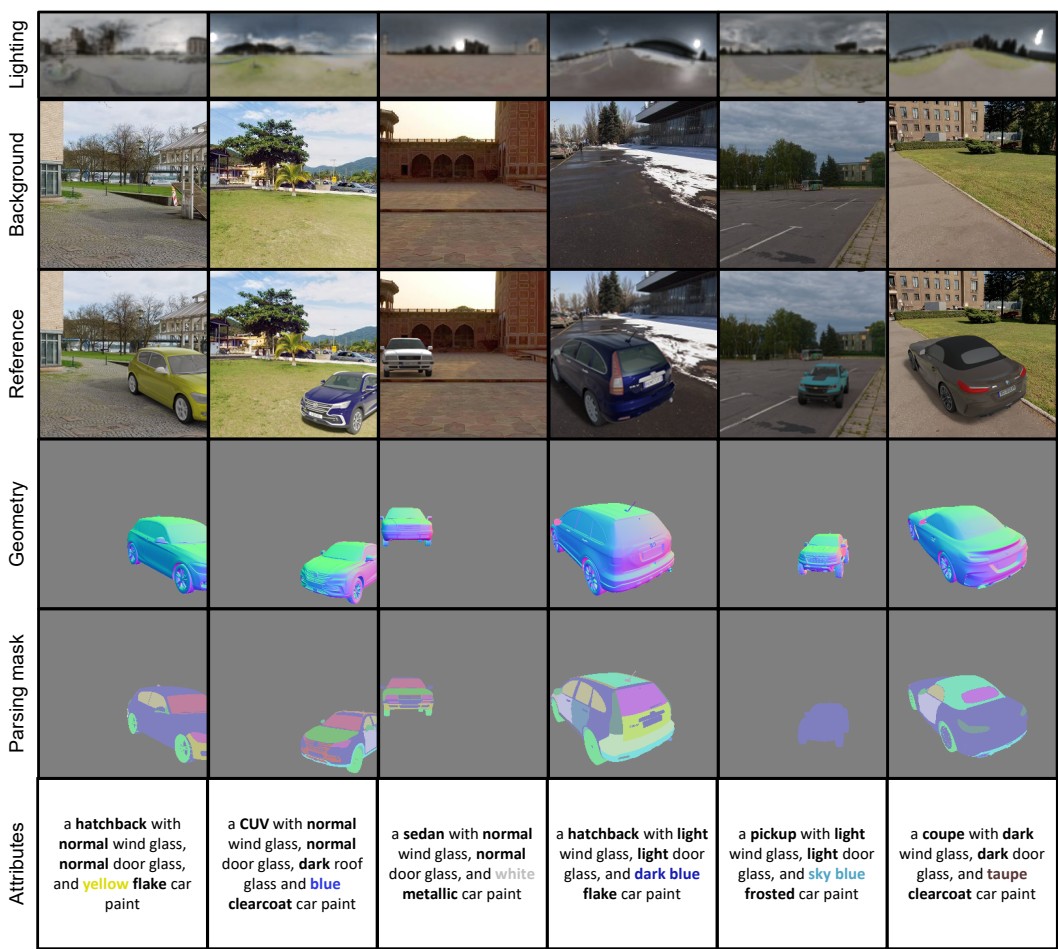

Figure 12: Data examples of our CAR-LUMINAIRE dataset.

dataset, we assure the images of each combination are only shown in one set. We only use images with the pixel ratio of the foreground between 10% and 50%, leaving 58,521 images (41,058 for the training set, 12,141 for the testing test, and 5,322 for the validation set). Each set of data contains a background image ($256 \times 256$), a lighting annotation ($128 \times 64$ envmap), a rendered reference image ($256 \times 256$), a geometry annotation ($256 \times 256$), a parsing mask ($256 \times 256$), and an attribute annotation. Here we give more data examples of our CAR-LUMINAIRE dataset in Fig. 12.

## 8.2 Details on the Parametric Lighting Representation

As stated in Sec. 5.1 of the main paper, we use a combination of 2-nd order SH lighting and directional lighting as our parametric lighting representation, where the low-frequency SH lighting is designed to fit the ambient lighting in the environment and the high-frequency directional lighting is used to describe the sunlight. Since the original lighting annotations in our CAR-LUMINAIRE dataset are envmaps, therefore, conversions have to be made to get the training labels in our parametric lighting representation.

The part of directional light is represented as $L_{\text{dir}} = \{z_{\text{vis}}, z_{\text{int}}, z_{\text{ang}}, c_{\text{sun}}, l_{\text{sun}}\}$ in our representation. For each envmap, if the maximum grey-sacle intensity $z_{\text{int}}$ is larger than a threshold $\delta_{\text{sun}} = 100$, the sun visibility $z_{\text{vis}}$ is set as 1 otherwise 0 (and other parameters treated as invalid). Then we use a manually set ratio $r_{\text{sun}} = 0.1$ and only keep pixels with grey-scale intensity larger than $r_{\text{sun}} z_{\text{int}}$. We calculate the intensity-weighted mass center of the connected area $\mathcal{A}_{\text{sun}}$ containing the pixel of maximum intensity, as the direction of the sun $l_{\text{sun}} \in \mathbb{R}^2$ in the spherical coordinates. The diameter

of $\mathcal{A}_{\text{sun}}$ in pixels is used as $z_{\text{ang}}$ and the mean RGB values divided by the mean intensity values over $\mathcal{A}_{\text{sun}}$ is used as the RGB weights $c_{\text{sun}} \in \mathbb{R}^3$.

The ideal directional light has no corresponding solid angle $\omega$ and thus can not be directly used in Eq. (10) of the main paper, we approximately "assign" a small solid angle $\omega_{\text{dir}}$ corresponding to a pixel (minimum visible unit) in the envmap. To calculate the equivalent intensity $i_{\text{dir}}$ corresponding to the pixel in the envmap, we assume the intensity from the sun center to the surroundings approximately fits the Gaussian distribution $f_{\text{G}}(x) = \alpha_{\text{G}} \exp\left(-\frac{x^2}{2\sigma_{\text{G}}^2}\right)$, and therefore we have:

$$f_{\text{G}}(0.5) = z_{\text{int}}, \quad f_{\text{G}}(z_{\text{ang}}/2) = r_{\text{sun}} z_{\text{int}}, \tag{11}$$

and we can solve $\alpha_{\text{G}} = z_{\text{int}} r_{\text{sun}}^{-\frac{1}{z_{\text{ang}}^2 - 1}}$ and $\sigma_{\text{G}} = \sqrt{\frac{z_{\text{ang}}^2 - 1}{8 \ln\left(1/r_{\text{sun}}\right)}}$ from Eq. (11) and therefore we have:

$$i_{\text{dir}} = \int_0^{z_{\text{ang}}/2} f_{\text{G}}(r) \mathrm{d}\omega(r) / \omega_{\text{dir}} \approx \int_0^{z_{\text{ang}}/2} f_{\text{G}}(r) 2\pi r \mathrm{d}r / \omega_{\text{dir}}. \tag{12}$$

We use 2-nd order SH coefficients $\sigma_{\text{SH}} = \{\sigma_{\text{SH}}^{\text{R}}, \sigma_{\text{SH}}^{\text{G}}, \sigma_{\text{SH}}^{\text{B}}\}$ to represent low-frequency light in each RGB color channel, where each $\sigma_{\text{SH}}^* = \{\sigma_{0,0}^*, \sigma_{1,-1}^*, \sigma_{1,0}^*, ..., \sigma_{2,2}^*\}$ are the corresponding coefficients for the 2-nd order spherical harmonics basis $\{Y_{0,0}, Y_{1,-1}, Y_{1,0}, ..., Y_{2,2}\}$. Due to the orthogonality of the spherical harmonics basis, the coefficients for low-frequency lighting $i_{\text{SH}}(\theta, \varphi)$ are computed as:

$$\sigma_{l,m}^* = \int_0^{2\pi} \int_0^{\pi} i_{\text{SH}}^*(\theta, \varphi) Y_{l,m}(\theta, \varphi) \sin\theta \mathrm{d}\theta \mathrm{d}\varphi, \tag{13}$$

where we use the envmap annotation (clipped into $[0, r_{\text{sun}} z_{\text{int}}]$ if $z_{\text{vis}}$ is 1) as $i_{\text{SH}}(\theta, \varphi)$. The reconstructed $\hat{i}_{\text{SH}}(\theta, \varphi)$ is simply the weighted sum of the spherical harmonics basis:

$$\hat{i}_{\text{SH}}^*(\theta, \varphi) = \sum_{i=0}^{l} \sum_{j=-l}^{l} \sigma_{i,j}^* Y_{i,j}(\theta, \varphi). \tag{14}$$

## 8.3 Details on the Illumination Image Rendering

The rendering of the illumination (candidate) images is conducted by applying Eq. (10) of the main paper pixel-wisely. Since the actual camera FoV is unknown, here we assume the camera viewing directions of all pixels are the same $v = (0, 0, -1)$ (orthogonal camera model), which is shown in Fig. 5 of the main paper to be a reasonable approximation for lighting-realistic generation tasks.

Since we use the normal map as the representation of geometry, which is not a complete 3D shape model (such as meshes, or signed distance functions), we only calculate single-bounce light effects, ignoring complex light transport effects such as self-cast shadow or inter-reflections. This is a trade-off between using the costly (and maybe more unreliable) single-view full 3D reconstruction or ignoring inconspicuous indirect light bounces.

The integration over the hemisphere $\Omega_n$ in Eq. (10) can be done discretely on an envmap. Therefore, the most intuitive way for the calculation is converting our parametric lighting representations back to envmaps before applying Eq. (10). However, a more efficient computation can be done utilizing the properties of our parametric representation, where we use $\rho \in \{1, 2, 4, 8, 16, 32, 64, 128\}$ for $\{I_c\}$.

Specifically, we divide $I$ as the sum of two parts $I_{\text{SH}}$ and $I_{\text{dir}}$ corresponding to our representation. Then each pixel $p$ of $I_{\text{diff,dir}}$ and $I_{\text{spec,dir}}^\rho$ can be calculated without integration as $i_{\text{dir}} c_{\text{sun}} (n_p \cdot l_{\text{dir}}) \omega_{\text{dir}}$ and $i_{\text{dir}} c_{\text{sun}} (n_p \cdot h_{\text{dir}})^\rho \omega_{\text{dir}}$, where $l_{\text{dir}}$ is the Cartesian coordinate representation of $l_{\text{sun}}$ and $h_{\text{dir}} = l_{\text{dir}} - v / ||l_{\text{dir}} - v||$ is the half vector introduced in Sec. 5.2 of the main paper. The negative dot product is clipped to 0 to avoid underflow. Besides, we also clip the minimums of $I_{\text{SH}}$ to 0.

For $I_{\text{diff,SH}}$, each pixel $p$ is fast calculated by using $Y_{l,m}(\theta, \varphi)$ to describe the distribution of $l$ [13]:

$$I_{\text{diff,SH}}^p = [c_1 \sigma_{2,2}(n_x^{p\,2} - n_y^{p\,2}) + c_3 \sigma_{2,0} n_z^{p\,2} + c_4 \sigma_{0,0} - c_5 \sigma_{2,0} \tag{15}$$
$$+ 2c_1(\sigma_{2,-2} n_x^p n_y^p + \sigma_{2,1} n_x^p n_z^p + \sigma_{2,-1} n_y^p n_z^p) + 2c_2(\sigma_{1,1} n_x^p + \sigma_{1,-1} n_y^p + \sigma_{1,0} n_z^p)]/\pi,$$

with weights $c_1 = 0.429043$, $c_2 = 0.511664$, $c_3 = 0.743125$, $c_4 = 0.886227$, and $c_5 = 0.247708$.

For $I_{\text{spec,SH}}^{\rho}$, we have $\theta_l = 2\theta_h$ and $\varphi_l = \varphi_h$. Similarly, $\hat{Y}_{l,m}(\theta, \varphi) = Y_{l,m}(2\theta, \varphi)$ is used to describe the distribution of $h$ [22], which gives the fast approximation of pixel $p$ with Blinn-Phong model [2]:

$$I_{\text{spec,SH}}^{\rho,p} \approx \{\sigma_{0,0}(c_4)^{\rho} + \sigma_{1,-1}(4c_2 n_y^p n_z^p)^{\rho} + \sigma_{1,0}[2c_2(2n_z^{p2} - 1)]^{\rho} + \sigma_{1,1}(4c_2 n_x^p n_z^p)^{\rho} \tag{16}$$
$$+ \sigma_{2,-2}(8c_1 n_x^p n_y^p n_z^{p2})^{\rho} + \sigma_{2,-1}[2c_1(4n_y^p n_z^{p3} - 2n_y^p n_z^p)]^{\rho} + \sigma_{2,0}[c_5(12n_z^{p4} - 12n_z^{p2} + 2)]^{\rho}$$
$$+ \sigma_{2,1}[2c_1(4n_x^p n_z^{p3} - 2n_x^p n_z^p)]^{\rho} + \sigma_{2,2}[c_1(4n_x^{p2} n_z^{p2} - 4n_y^{p2} n_z^{p2})]^{\rho}\}(\rho + 4)/8\pi.$$

## 8.4   Details on the User Study

We randomly sample 200 sets of results of compared methods and ask volunteers to choose one in each set that best matches the following description: *(i)* "The repainted region which seems most *realistic*"; *(ii)* "the whole repainted image which seems most *harmonized* in lighting"; *(iii)* "the whole repainted image which seems most *realistic overall*".

The volunteers are shown with the masked repainted foreground images, *i.e.*, without the background context when asked about the *realistic* question. Then the full repainted images are shown and the *harmonized* question is asked on the same set of results, where we use our repainted background region for all results to prevent our method to be identified or guessed out by only noticing the difference in the background. The original results of compared methods are shown to the volunteers when asking about the *realistic overall* question. We first ask the *realistic* question, then the *harmonized* question, and at last the *realistic overall* question. We have reported the results of the first two questions in Tab. 1 of the main paper while the results for the *realistic overall* question are: **Ours**: **77.32%**, **Pavllo *et al*.** [12]: 14.50%, **UniCoRN** [6]: 7.03%, **Weng *et al*.** [18]: 0.92%, **MISC** [19]: 0.23%.

The order of sets and images in each set is randomized, and we deliberately duplicate 5 sets of the samples as the quality control questions to judge whether the volunteers have paid attention when finishing the questionnaires. Questionnaires that failed in the quality control questions are discarded.

## 8.5   Training Details

**Experimental settings.** Our pipeline is implemented in PyTorch [11] and trained step-wise. We first train our *NetL* on the held-out background images with a batch size of 64 and an initial learning rate of $1 \times 10^{-4}$ (which halves every 20 epochs) for 60 epochs, where we estimate the sun position $l_{\text{sun}}$ in the form of an $8 \times 32$ classification task and we apply log-compressed tone mapping [8] $T = {}^{\log(1+16H)}/_{\log(1+16)}$ for the HDR sun intensity $z_{\text{int}}$. Our *NetS* and *NetB* are separately trained on our CAR-LUMINAIRE dataset with a batch size of 32 and a fixed learning rate of $2 \times 10^{-4}$ for 60 epochs. Then we run our full pipeline optimization (one discriminator step after each generator step) with fixed *NetL*, *NetS*, and *NetB* to learn the network parameters of *NetF*, with a batch size of 24 and a fixed learning rate of $2 \times 10^{-4}$ for 30 epochs. During the training of *NetS* and *NetF*, we use the hierarchical labeling enhancement at the probability of 0.5, where each part label has a probability of 0.5 to be coarsened to its upper-level label. Before illumination injection, the illumination image $I$ is clipped by an empirically set threshold $\delta_I = 2.0$ to simulate the over-exposure of highlights in LDR images and avoid extremely high inputs to network layers. For cross-modality conditional consistency constraints, we pretrain the image encoder $\text{Enc}^i$ (omitted in the main paper) and the attribute encoder $\text{Enc}^c$ (Fig. 3) on our CAR-LUMINAIRE dataset following previous work [21].

The baseline methods are trained on our CAR-LUMINAIRE dataset with the same batch size of 24 as our *NetF* for 30 epochs using their default settings in their released code. We use Adam optimizer [9] in all of our experiments, and all experiments are conducted on 4 NVIDIA Tesla V100 graphic cards.

**Training losses.** Our full pipeline is trained with the following losses:

$$\mathcal{L} = \mathcal{L}_{\text{L}} + \mathcal{L}_{\text{S}} + \mathcal{L}_{\text{B}} + \mathcal{L}_{\text{F}}, \tag{17}$$

where $\mathcal{L}_{\text{L}}$, $\mathcal{L}_{\text{S}}$, $\mathcal{L}_{\text{B}}$, and $\mathcal{L}_{\text{F}}$ are the loss terms for our *NetL*, *NetS*, *NetB*, and *NetF*, respectively.

For our *NetL*, $\mathcal{L}_{\text{L}}$ consists of two parts $\mathcal{L}_{\text{L}} = \mathcal{L}_{\text{SH}} + \mathcal{L}_{\text{dir}}$ corresponding to our lighting modeling:

$$\mathcal{L}_{\text{SH}} = \mathcal{L}_{\text{coeff}} + \mathcal{L}_{\text{pano}}, \ \ \mathcal{L}_{\text{dir}} = \mathcal{L}_{\text{vis}} + \mathcal{L}_{\text{pos}} + \mathcal{L}_{\text{param}}, \tag{18}$$

where $\mathcal{L}_{\text{coeff}}$ is an $L_2$ loss for $\sigma_{\text{SH}}$ with $\sigma_{l,m}^*$ from Eq. (13), $\mathcal{L}_{\text{pano}}$ is an $L_1$ loss for envmaps reconstructed by SH coefficients $\hat{i}_{\text{SH}}^*(\theta, \varphi)$ from Eq. (14) with $i_{\text{SH}}(\theta, \varphi)$, $\mathcal{L}_{\text{vis}}$ is a binary cross-entropy loss

for $z_{\text{vis}}$, $\mathcal{L}_{\text{pos}}$ is a cross-entropy loss for the $8 \times 32$ classification results of $l_{\text{sun}}$, and $\mathcal{L}_{\text{param}}$ are $L_2$ losses for the remaining parameters (log-compressed $z_{\text{int}}$, $z_{\text{ang}}$, and $c_{\text{sun}}$). For images with the sun not visible ($z_{\text{vis}} = 0$) in the lighting annotations, we set $\mathcal{L}_{\text{pos}} = \mathcal{L}_{\text{param}} = 0$.

For our *NetS*, $\mathcal{L}_{\text{S}}$ is defined as:
$$\mathcal{L}_{\text{S}} = \mathcal{L}_{\text{sp}} + \mathcal{L}_{\text{s-smooth}}, \tag{19}$$
where $\mathcal{L}_{\text{sp}}$ is an $L_2$ loss for $G$, and $\mathcal{L}_{\text{s-smooth}} = \sum [(\nabla_i G)^2 + (\nabla_j G)^2]$ is the smoothness loss for $G$.

For our *NetB*, $\mathcal{L}_{\text{B}}$ is defined as:
$$\mathcal{L}_{\text{B}} = \mathcal{L}_{\text{bg}} + \mathcal{L}_{\text{b-smooth}} + \mathcal{L}_{\text{b-dis}}, \tag{20}$$
where $\mathcal{L}_{\text{bg}}$ is an $L_1$ loss for $y^{\text{b}}$, $\mathcal{L}_{\text{b-smooth}} = \sum [(\nabla_i (y^{\text{b}}/x^{\text{b}}))^2 + (\nabla_j (y^{\text{b}}/x^{\text{b}}))^2]$ is the smoothness loss for $y^{\text{b}}$, and $\mathcal{L}_{\text{b-dis}}$ is the discriminator loss for $y^{\text{b}}$ with the background regions of reference images.

For our *NetF*, $\mathcal{L}_{\text{F}}$ is defined following UniCoRN [14] as:
$$\mathcal{L}_{\text{F}} = \mathcal{L}_{\text{fg}} + \mathcal{L}_{\text{r}} + \mathcal{L}_{\text{bc}} + \mathcal{L}_{\text{fm}} + \mathcal{L}_{\text{per}} + \mathcal{L}_{\text{cm}}, \tag{21}$$
where $\mathcal{L}_{\text{fg}}$ and $\mathcal{L}_{\text{r}}$ are discriminator losses judging whether $y^{\text{f}}$ is real and whether $y^{\text{r}}$ is composited, $\mathcal{L}_{\text{bc}}$ is an $L_1$ loss enforcing $(1 - m) \odot y^{\text{f}}$ close to $x^{\text{b}}$, $\mathcal{L}_{\text{fm}}$ and $\mathcal{L}_{\text{per}}$ are the feature matching loss [17] and the perceptual loss [3] for $y^{\text{r}}$, and $\mathcal{L}_{\text{cm}}$ is the cross-modality conditional matching loss [21].

## 8.6 Implementations of Baseline Methods

We use the released code of UniCoRN [14], MISC [19], Weng *et al.* [18], and Pavllo *et al.* [12] as the implementations of our baseline method. As mentioned in the main paper, modifications have been made to the released code of Weng *et al.* [18] and Pavllo *et al.* [12] for taking attributes as the input color condition $x^{\text{c}}$. Besides, for Pavllo *et al.* [12], since their generated background is not conditioned on either the original background or other conditions and thus is not controllable, we discard their generated background and replace it with the input background image to fit the formulation of CIR.

## 8.7 Additional Results

**Lighting and shape estimations.** Our LuminAIRe pipeline consists of lighting and shape estimations which will inevitably introduce errors. As stated in Sec. 5.1 of the main paper, the low-frequency part (SH lighting) and high-frequency part (directional light) of the lighting are separately estimated. Here we report the lighting estimation errors from masked background images (foreground regions masked by zeros) in each part: *(i)* directional (sun) light: mean angular error (MAE): 28.37°, mean azimuth error: 3.84°, and mean elevation error: 27.74°; *(ii)* SH lighting: mean absolute error of SH coefficients: 0.0488, mean square error of SH coefficients: 0.0054, mean absolute error of envmaps reconstructed by SH coefficients: 0.0435, and mean square error of envmaps reconstructed by SH coefficients: 0.0043. Similarly, we report the estimation errors on normalized normal maps in the shape estimation: mean angular error (MAE): 9.83°, mean absolute error: 0.0167, and mean square error: 0.0039. These errors would prevent us from recovering the exact lighting effects, however, are tolerable for the demand of lighting-realistic repaintings.

**More comparisons and ablation variants.** We conduct a breakdown evaluation on how our method and compared methods work on foreground regions (noted as "fg.") and how repainted background regions (noted as "bg.") by our method contribute to the realistic perception. We also compare our method with more ablation variants (**Ours-A** and **Ours-AI**) for the completeness of the ablation study. Despite that we can not compare our method with image harmonization methods in an exact fair setting, as an intuitive reference, we choose two of the latest methods (**DHT+** [5] and **PCT-Net** [4]) and use the repainting results from **Ours-AI** as their inputs. The quantitative results are shown in Tab. 2 and the qualitative comparisons are shown in Fig. 13, where the harmonized images show better integrity than input as M-score indicates, however, do not show better lighting effects and may have severe color-shifting issues as R-prcn and SSIM scores indicate.

**More results on our** CAR-LUMINAIRE **dataset.** More qualitative results on our CAR-LUMINAIRE dataset are shown in Fig. 14 and Fig. 15. Our LuminAIRe generally performs better qualitatively than baseline methods in generating realistic, harmonized, and consistent lighting effects.

**In-the-wild performance.** To test the generalization ability of the compared methods, we show qualitative results of in-the-wild data which are collected from the Waymo dataset [15] and the

Table 2: Additional quantitative evaluation results. Separated evaluations of foreground and background regions are shown. Qualitative results of additional ablation variants and image harmonization baselines are also shown. ↑ (↓) means higher (lower) is better. "fg." stands for "foreground" and "bg." stands for "background". Please note that "original bg." corresponds to the background regions of all compared baseline methods since they leave the background untouched.

| Method | FID ↓ | R-prcn ↑ | M-score ↓ | SSIM ↑ |
|---|---|---|---|---|
| MISC fg. | 76.26 | — | — | 0.8228 |
| Weng *et al.* fg. | 44.14 | — | — | 0.8306 |
| Pavllo *et al.* fg. | 6.14 | — | — | 0.8671 |
| UniCoRN fg. | 9.53 | — | — | 0.8541 |
| Ours fg. | **4.30** | — | — | **0.8689** |
| Original bg. | 21.43 | — | — | 0.8309 |
| Ours bg. | **4.94** | — | — | **0.8494** |
| Ours-A | **5.04** | **74.29%** | 13.76 | **0.7167** |
| Ours-AI | 5.72 | **74.73%** | 15.36 | 0.7106 |
| DHT+ [5] | 5.94 | 67.34% | 9.02 | 0.7057 |
| PCT-Net [4] | 5.31 | 69.59% | **7.85** | 0.7035 |

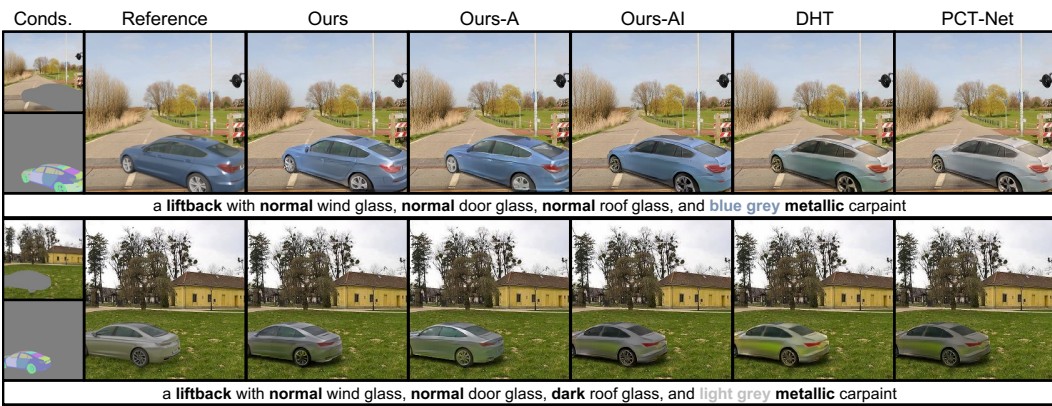

Figure 13: Qualitative results of additional ablation variants and image harmonization baselines.

UASOL dataset [1] in Fig. 16. Although their data distribution is far different from our synthetic data, our LuminAIRe still gives reasonable lighting-realistic results compared with baseline methods.

**Failure cases.** Here we analyze examples of failure cases in Fig. 17. When the repainted region is across the boundary of the shadows (the first row), the global lighting assumption may lead to unrealistic lighting effects. A too-coarse parsing mask (the second row) would raise serious geometry ambiguity and renders a failed repainting. The lighting effects would become less realistic if the accumulated errors in lighting and shape estimations were too large (the third row). The occasionally badly repainted background (the fourth row) would also do harm to the lighting-realistic perception.

## 8.8   Detailed Network Architectures

We show the detailed network architectures of the *NetL*, *NetS*, *NetB*, and *NetF* from Fig. 19 to Fig. 21, with the structures and default settings of common blocks shown in Fig. 18.

The network architectures of the image encoder $Enc^i$ and the attribute encoder $Enc^c$ for measuring cross-modality conditional consistency remain the same with the HCMSM proposed in UniCoRN [14]. We adopt the network backbone of their $F^G$ for our *NetF*, where we inject the illumination images $I$ as the illumination condition $x^i$ in 2D image space from the resolutions of $32 \times 32$ to $256 \times 256$. Specifically, we replace the batch normalization layers with instance normalization layers in FABN module and ignore the texture condition $x^p$ when injecting the illumination image $I$.

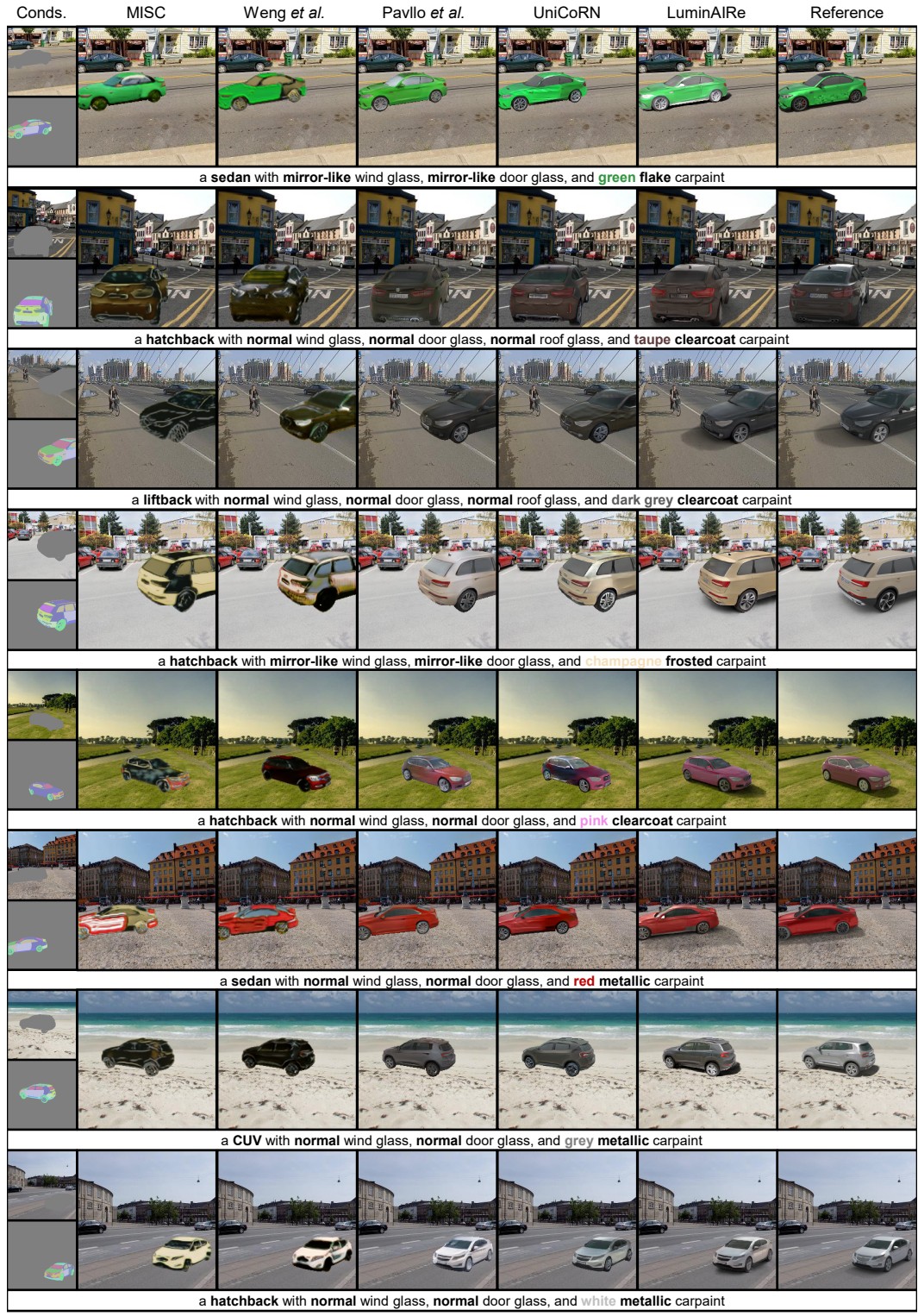

Figure 14: More qualitative comparisons on our CAR-LUMINAIRE dataset.

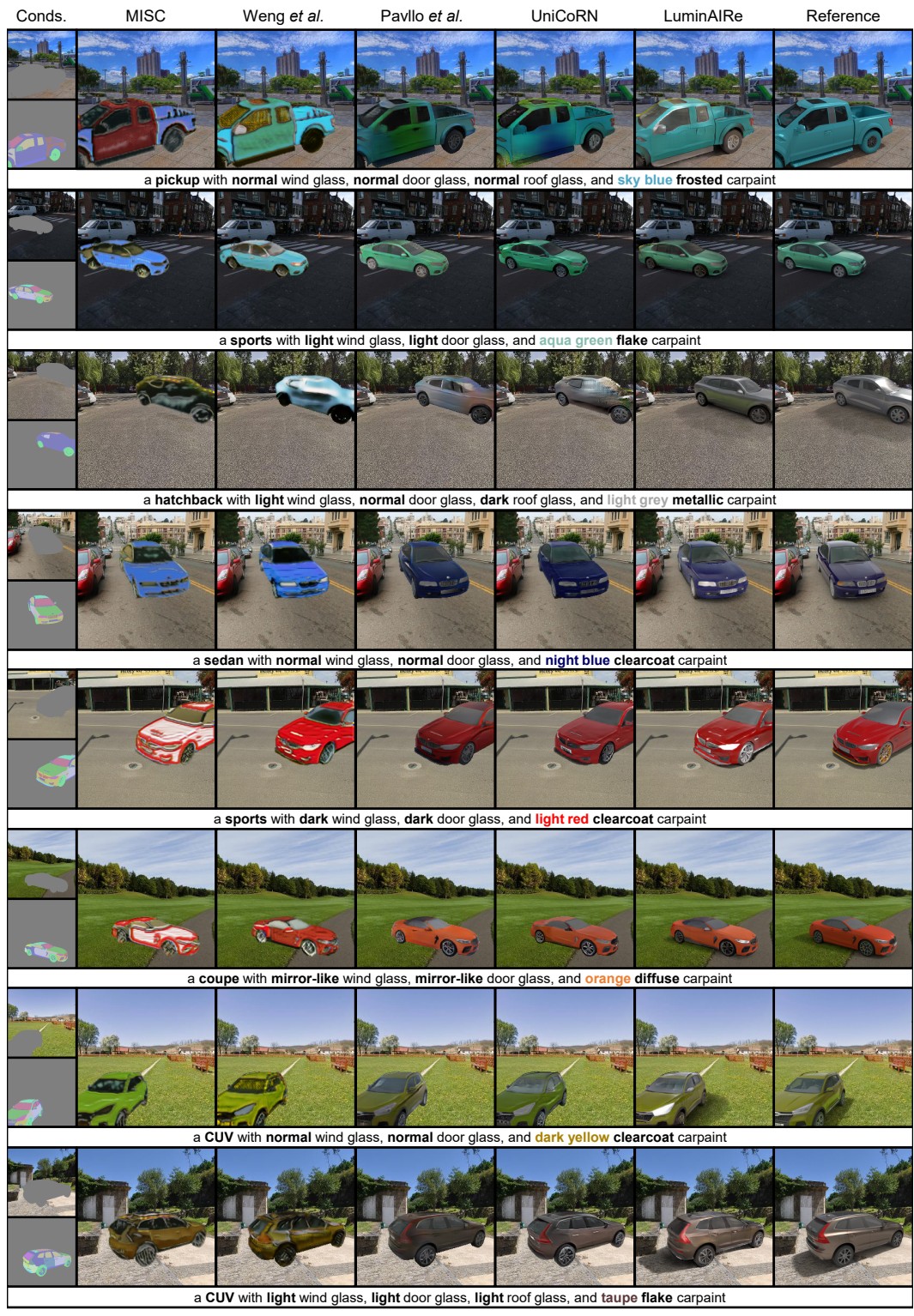

Figure 15: More qualitative comparisons on our CAR-LUMINAIRE dataset.

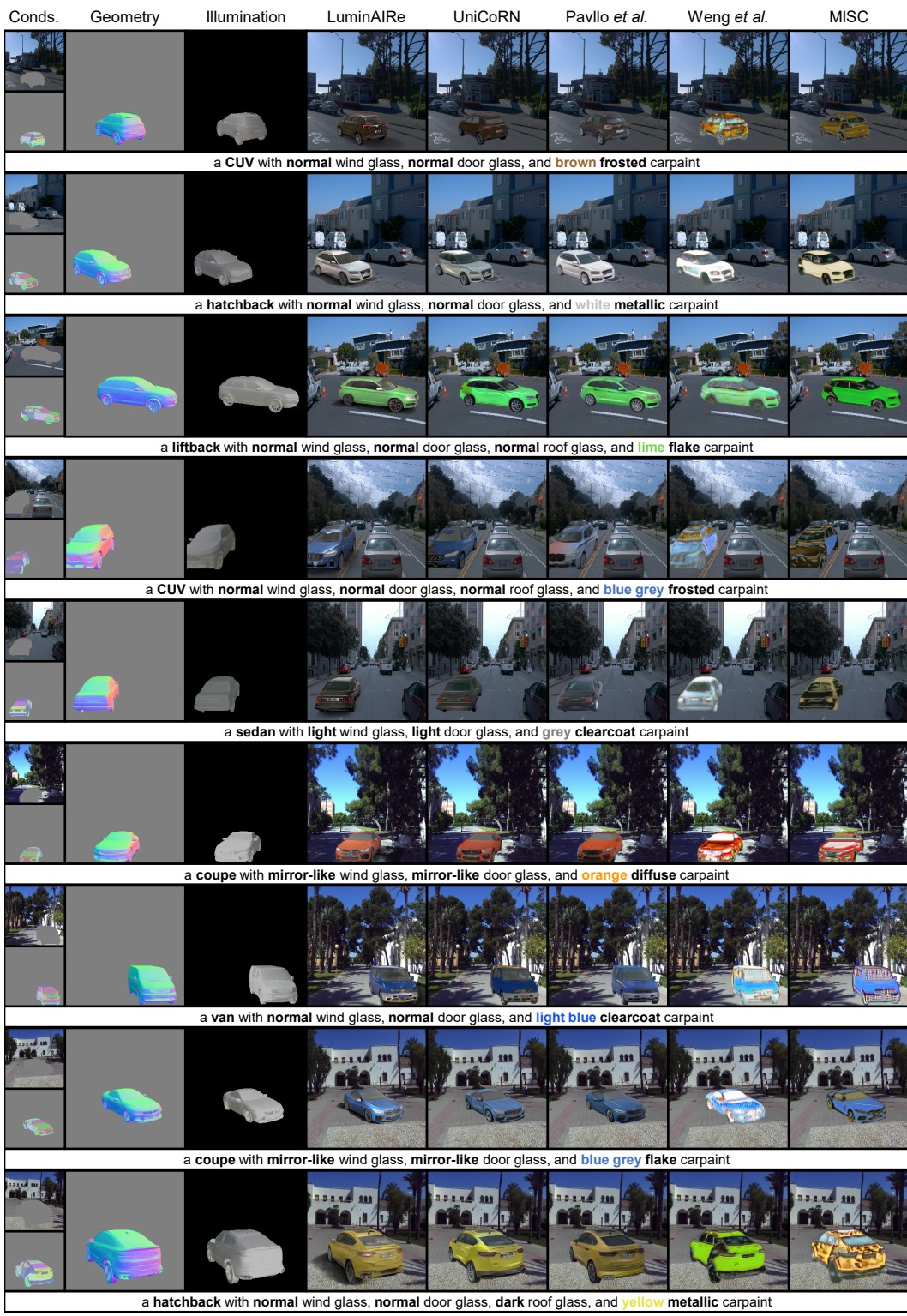

Figure 16: More qualitative comparisons on in-the-wild data.

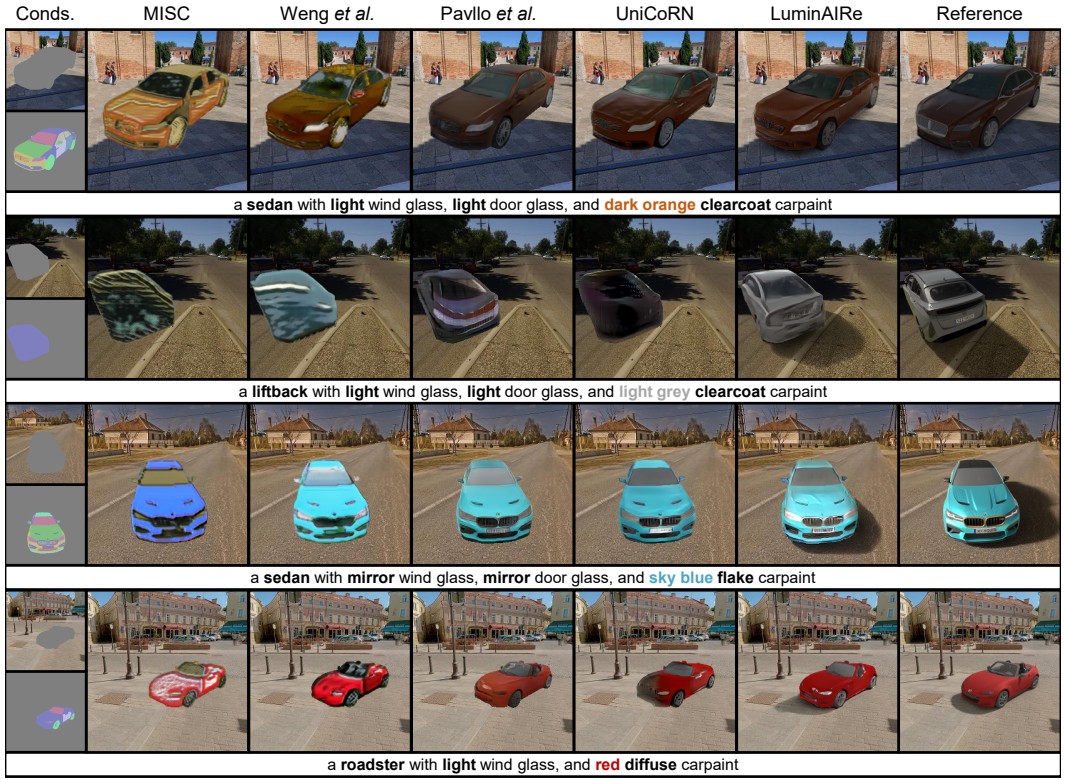

Figure 17: Failure cases on our CAR-LUMINAIRE dataset.

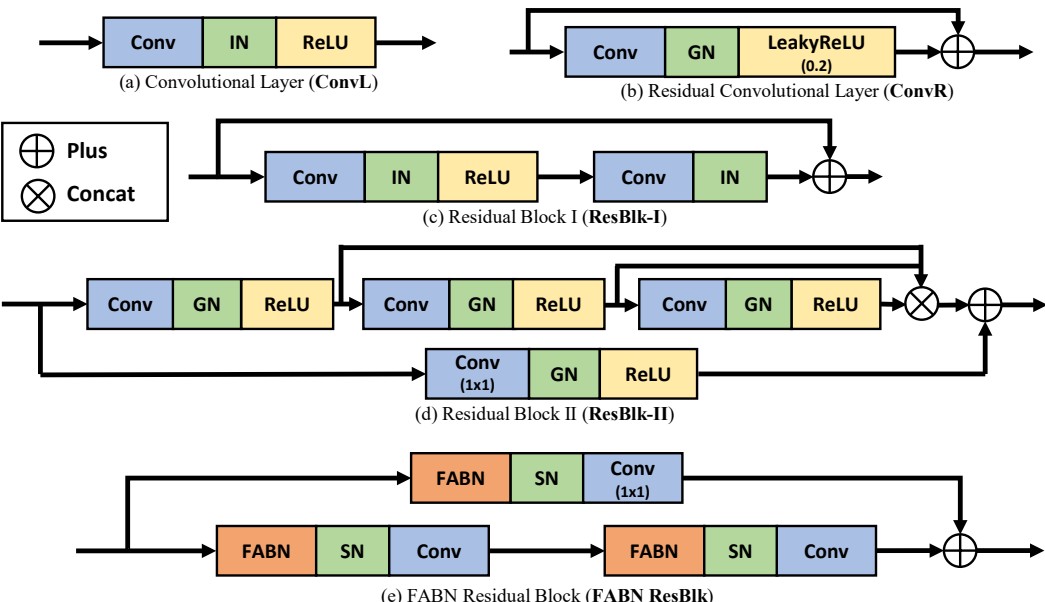

Figure 18: Common blocks used in the network architectures. Notations: BN = Batch Normalization [7], IN = Instance Normalization [16], GN = Group Normalization [20], SN = Spectral Normalization [10], FABN = Feature Adaptive Batch Normalization [14].

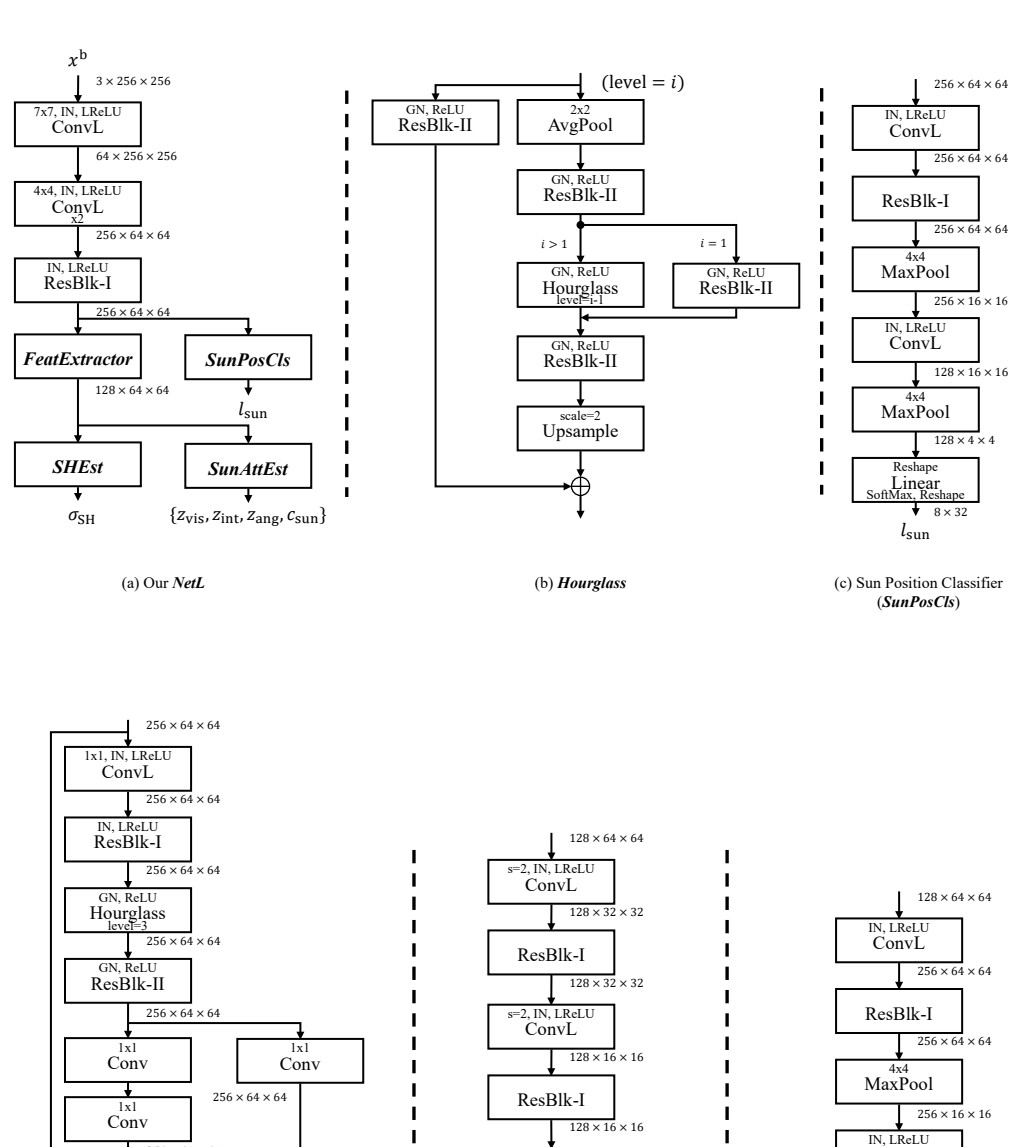

Figure 19: Network architectures of our pipeline.

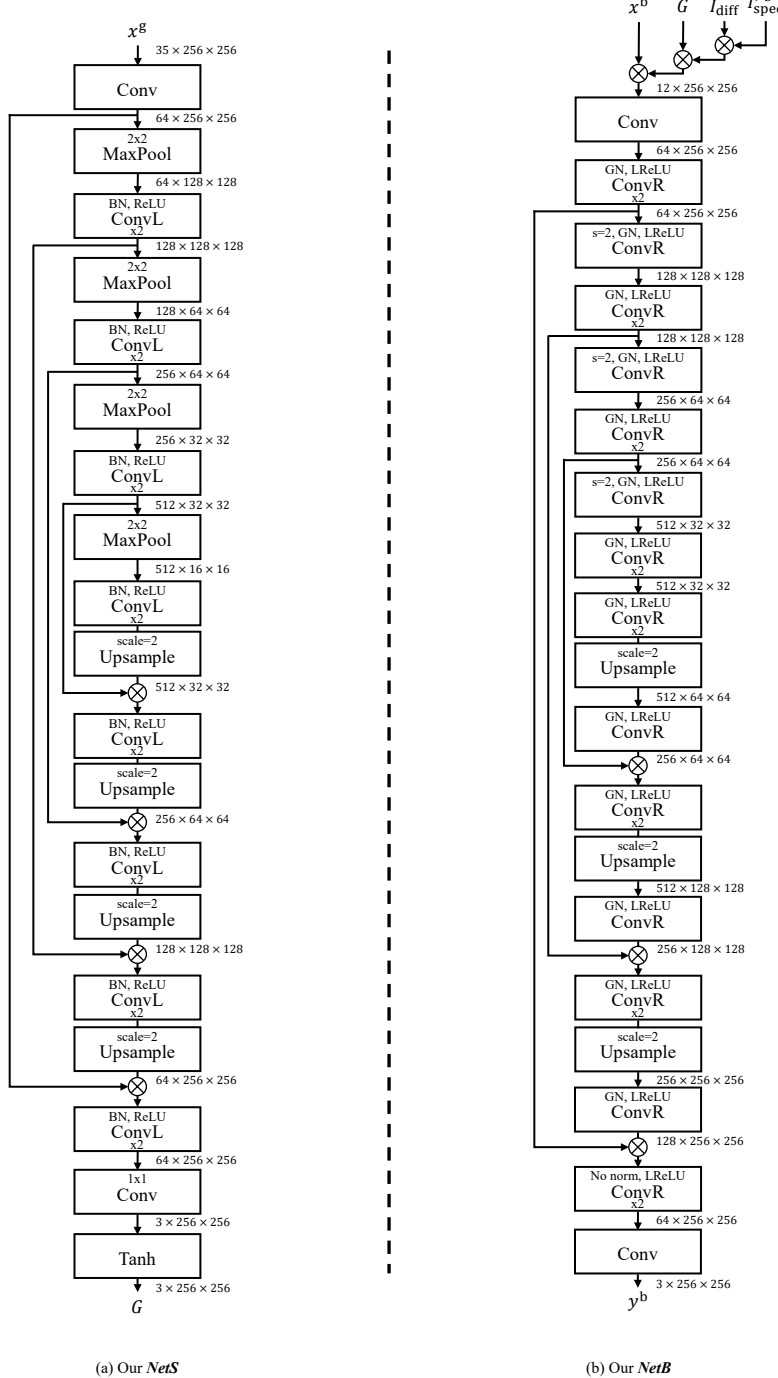

(a) Our *NetS*

(b) Our *NetB*

Figure 20: Network architectures of our pipeline (cont'd).

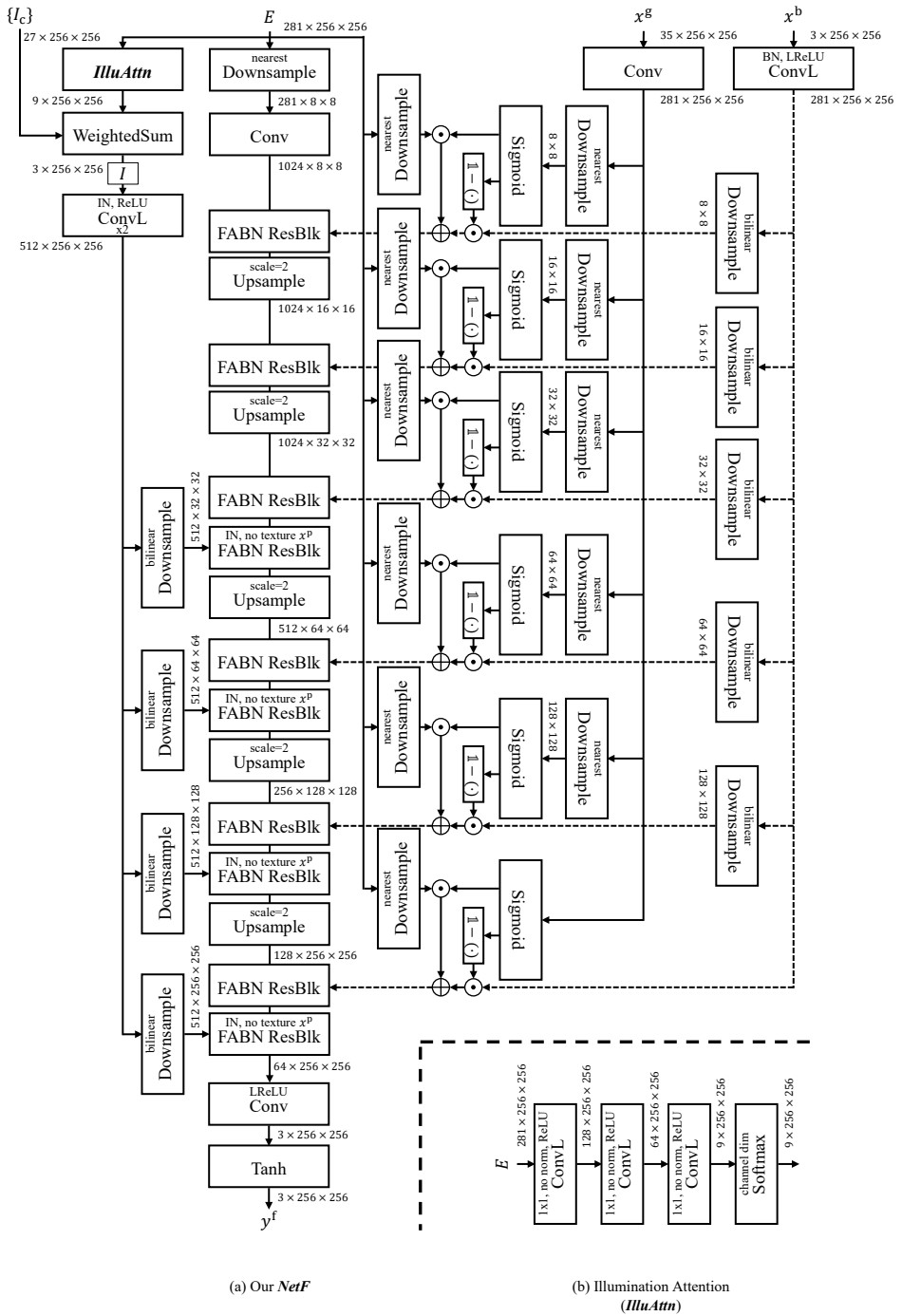

(a) Our *NetF*

(b) Illumination Attention
(*IlluAttn*)

Figure 21: Network architectures of our pipeline (cont'd).