# OpenReview forum: "LuminAIRe: Illumination-Aware Conditional Image Repainting for Lighting-Realistic Generation"
_NeurIPS.cc/2023/Conference — NeurIPS 2023 poster_

### Official Review · Reviewer_Kvqo · 2023-07-02

**Soundness:** 3 good
**Presentation:** 4 excellent
**Contribution:** 3 good
**Rating:** 5
**Confidence:** 3

**Summary:**

This paper proposes a method for illumination-aware conditional image repainting (LumiAIRe).
Different from conventional conditional image repainting (CIE), LumiAIRe combines environmental lighting estimation, 3D normal estimation, and illumination injection for achieving harmonized lighting effects in both foreground and background regions.
To validate the effectiveness of the proposed illumination-aware repainting method, a new dataset named CAR-LumiAIRe with 52581 composited images using 198 detailed 3D car models and 1321 background images was proposed.
The overall performance of LumiAIRe is convincing especially on the generated shadows in the background images and the illumination of the car in the foreground images.
The effectiveness of the proposed method was well validated by the experiments and ablation studies.

**Strengths:**

+ Combining physical information into vision tasks is a promising and practical direction. The idea of incorporating lighting and 3D geometric information for illumination-aware conditional image repainting is interesting and sound.
+ The repainted car images are with more plausible lighting effects than previous methods.
+ The problem formulation is clear and also keeps consistent with the equations in Sec.5, which is easy to follow and understand.
+ For dataset creation, the insertion points are carefully specified by segmenting the "placeable flat ground", which is more physically plausible than previous datasets.

**Weaknesses:**

- The variation of the shape of the car in the results and experiments is too small. This means that the shape of the repainted cars is too similar to each other, which may limit the shape generalization capacity of the proposed method.
- CAR-LumiAIRe was rendered at the resolution of 256x256, which is not enough to support high-resolution CIE tasks.

**Questions:**

- Why not also estimate the 3D shape and position of the car for direct relighting without using so many neural networks?
- In Fig.7, the authors claim that "the illumination injection helps foreground generation by comparing ours-HA and ours-HAI". However, although missing some sort of lighting effects, in my opinion, the results of ours-HAI look better than ours-HA, please clarify this in the response.
- How about other types of objects? Adding a type of object with more significant shape variation than cars will significantly improve the quality of the paper, but will clearly take much more effort, so this is only an option but not serious advice.

**Limitations:**

Yes.

---

> ### Author Rebuttal · Authors · 2023-08-09
>
> Thanks for your very detailed review and suggestions, and here we give responses to the mentioned concerns.  We are looking forward to discussing with you during the author-reviewer discussion period.
>
> ### Concerns about data diversity
>
> We have tried our best to collect car models with sufficient diversities. However, most available car models are biased in shape (see dataset statistics in the response for **reviewer oBS5**). This issue could be solved by introducing other categories of data which is less biased in shape distribution.
>
> ### Concerns about the resolution of the proposed dataset
>
> We use the low-resolution version ($256\times 256$) of CAR-LuminAIRe dataset in all of our experiments mainly to save computation resources and accommodate the memory limit of the GPU. We have also rendered a high-resolution version ($512\times512$) of the CAR-LuminAIRe dataset, which would be released upon acceptance to facilitate high-resolution tasks better.
>
> ### Discussion about potential full 3D solutions
>
> > Why not also estimate the 3D shape and position of the car for direct relighting without using so many neural networks?
>
> The motivation for our work starts from the image-level repainting tasks, and the finally desired output is in image space, which is user-friendly for people without expert skills. Therefore, the full 3D pipeline is unnecessary in our task setting. If we adopt the full 3D generation and pose estimation for the car, then we still need explicit lighting estimation for relighting. Besides, to generate reasonable shadows from 3D foreground objects, a (partial) 3D reconstruction of background scenes may also be needed (otherwise, we are back to using neural networks to generate shadows, which might be more complicated since the inputs are in 3D), which would be rather challenging considering the single image setting of our task. Our pipeline breaks down the LuminAIRe task into multiple easier sub-problems using different networks (all have simple architectures). However, a full 3D solution may be more challenging for networks to handle (*e.g.*, how to map the user-specified conditions onto the 3D relightable models), with more expensive computations for 3D processing.
>
> ### Discussion on visual qualities in the ablation study
>
> > However, although missing some sort of lighting effects, in my opinion, the results of ours-HAI look better than ours-HA.
>
> The main reason hurting the visual quality of **Ours-HA** may be the part of the car body where the color is not correctly generated, which corresponds to the “unspecified car part” semantic, as the parsing mask shows. Without hierarchical labeling enhancement in training, the trained models sometimes may wrongly apply/not apply a “strange” color to this part. For the specific case (the top row of **Fig. 7** of the main paper), **Ours-HAI** happens to apply a color close to (darker) the car paint color and may look better than **Ours-HA** taken when ignoring lighting effects (a similar effect is also observed for **Ours-H**). What we want to emphasize by that case is, without illumination injection, the lighting effects are rather unrealistic, leading to a much darker blue color and window glass. Statistically speaking, **Ours-HA** should perform better than **Ours-HAI** in terms of faithfully following the user-specified conditions (R-prcn and SSIM) and realism (FID and M-score), as indicated by the scores in **Tab. 1** of the main paper. A similar trend can also be observed from the first row of **Fig. II** in the attached pdf, where **Ours-A** demonstrates better lighting effects than **Ours-AI**, showing the effectiveness of the illumination injection.
>
> ### Scalability of the proposed method
>
> Please refer to **Sec. II** of the global response.

---

> ### Comment · Reviewer_oBS5 · 2023-08-17
>
> I would like to thank the authors for their efforts and responses. Although the authors provide a general explanation for the generalization of the proposed method, I still cannot be convinced until it is further validated. I think that this paper still needs to be throughtly polished and experimentally validated.  The paper currently does not meet the standards accepted by NeurIPS 2023, and I still keep on rejecting this paper.

---

> > ### Comment · Reviewer_Kvqo · 2023-08-18
> >
> > Dear Reviewer oBS5, it looks like you post your response to a wrong place :)

---

### Official Review · Reviewer_oBS5 · 2023-07-04

**Soundness:** 2 fair
**Presentation:** 2 fair
**Contribution:** 3 good
**Rating:** 4
**Confidence:** 5

**Summary:**

This paper presents the ilLumination-Aware conditional Image Repainting (LuminAIRe) task to address the unrealistic lighting effects based on recent conditional image repainting (CIR) methods. The main contributions include : 1) introducing a new task of ilLumination-Aware conditional Image Repainting (LuminAIRe), 2) designing a full LuminAIRe pipeline to acquire more realistically repainted results, and 3) collecting a new dataset CAR-LUMINAIRE with rich material and lighting condition variants. This paper is generally written and has a clear layout.

**Strengths:**

This paper main presents the ilLumination-Aware conditional Image Repainting (LuminAIRe) task to address the unrealistic lighting effects based on recent conditional image repainting (CIR) methods. To this end, the authors introduce a new task of ilLumination-Aware conditional Image Repainting (LuminAIRe), 2) designs a full LuminAIRe pipeline to acquire more realistically repainted results, and 3) collects a new dataset CAR-LUMINAIRE. This task is interesting, and the constructed dataset is useful for illumination community.

**Weaknesses:**

The work collected a new dataset CAR-LUMINAIRE, which is interesting and useful for the illumination community. However, in the process of collecting the dataset, how to perform the "warping" to achieve aligned envmap? How to consider the scale of foreground objects? I think it is necessary to conduct a statistical analysis of the synthesized dataset.

When estimating background illumination information, how to address the impact of the content occluded by foreground objects on the overall illumination information estimation? Especially for larger foreground objects?

Compared to the image harmonization task, this paper considers the shadow generation of foreground objects. How do the authors evaluate the quality of generated shadows?

The variety of foreground objects in the dataset is extremely limited, which greatly limits the application of the method. More types of foreground objects may be more convincing and improve the robustness of the method.






**Questions:**

see the Weakness

**Limitations:**

see the Weakness

---

> ### Author Rebuttal · Authors · 2023-08-09
>
> Thanks for your very detailed review and suggestions, and here we give responses to the mentioned concerns.  We are looking forward to discussing with you during the author-reviewer discussion period.
>
> ### More details on dataset creation
>
> > How to perform the "warping" to achieve aligned envmap?
>
> Since we adopt the spatially-uniform global lighting assumption for the outdoor scenes, we do not consider the depth when warping the environment map, and the warping is essentially a rotation that makes camera coordination of the virtual camera aligned with the coordination of the environment map (*i.e.*, making the virtual camera points toward the center of the aligned environment map and the up direction of the virtual camera is also the up direction of the aligned environment map). The rotation can be derived from the camera pitch and yaw (we use no camera roll to keep the ground in the cropped images level).
>
> > How to consider the scale of foreground objects?
>
> As stated in **L. 158-159** of the main paper,  all car models are resized to fit their real-world dimensions. Therefore, the scales of foreground objects appearing in the image are decided by the camera FoV used,  the 2D insertion point, and the accuracy of the off-the-shelf normal and depth estimation results (influence the conversion from the 2D insertion point to 3D relative position calculation).
>
> By converting the depth estimation results into real-world units, ideally (if all estimations are perfect), the rendered cars would be precisely the same size as a real car at the corresponding 3D point in the scene. However, due to estimation errors, the scale of the rendered objects may not always be reasonable, and therefore filtering of rendered data is conducted.
>
> > I think it is necessary to conduct a statistical analysis of the synthesized dataset.
>
> Thanks for the helpful suggestion. We follow it to provide more statistical analysis results of our Car-LuminAIRe dataset:
>
> 1. Portions of pixels of foreground region.
>
>    | 10%~15% | 15%~20% | 20%~25% | 25%~30% | 30%~35% | 35%~40% | 40%~45% | 45%~50% |
>    | :-----: | :-----: | :-----: | :-----: | :-----: | :-----: | :-----: | :-----: |
>    |  27.9%  |  21.2%  |  16.4%  |  12.0%  |  8.4%   |  5.6%   |  3.7%   |  4.8%   |
>
> 2. Distributions of the car types.
>
>    | hatchback | sedan | race | liftback |  CUV  | pickup | micro | roadster |
>    | :-------: | :---: | :--: | :------: | :---: | :----: | :---: | :------: |
>    |   14.6%   | 22.0% | 6.0% |   8.8%   | 14.0% |  5.3%  | 3.6%  |   2.7%   |
>
>    | SUV  | MPV  | minivan | sports | coupe | universal | minibus | super | van  |
>    | :--: | :--: | :-----: | :----: | :---: | :-------: | :-----: | :---: | :--: |
>    | 6.7% | 1.8% |  0.9%   |  3.8%  | 3.5%  |   2.3%    |  0.2%   | 2.5%  | 1.3% |
>
> 3. Distributions of the types of car paints.
>
>    | metallic | clearcoat | frosted | flake | diffuse |
>    | :------: | :-------: | :-----: | :---: | :-----: |
>    |  18.2%   |   26.8%   |  22.9%  | 19.8% |  12.3%  |
>
> 4. Distributions of the used camera FoVs.
>
>    | $27\degree\sim31\degree$ | $31\degree\sim36\degree$ | $36\degree\sim41\degree$ | $41\degree\sim46\degree$ | $46\degree\sim51\degree$ | $51\degree\sim56\degree$ | $57\degree\sim61\degree$ | $61\degree\sim66\degree$ |
>    | :----------------------: | :----------------------: | :----------------------: | :----------------------: | :----------------------: | :----------------------: | :----------------------: | :----------------------: |
>    |           3.1%           |           4.3%           |           8.9%           |          21.2%           |          23.0%           |          22.1%           |          11.9%           |           5.5%           |
>
> ### Concerns about lighting estimation
>
> As shown in **Fig. 3** of the main paper, our lighting estimation network (NetL) only takes the background image as input. The foreground region of the background image is masked with zeros. Thus there are no foreground objects in the background images distracting the lighting estimation process. Of course, having fewer informative pixels imposes a challenge,  we address this issue through three designs:
>
> 1. The NetL is trained with randomly chosen foreground masks applied on contact background images, forcing the network to have more robustness to occlusion (sorry for omitting the detail in the supplemental material).
> 2. The lighting representation we adopt is relatively less complex, which lowers the difficulty of getting a reasonable lighting estimation from limited pixel observations.
> 3. The synthetic dataset is filtered wrt. the portion of foreground pixels (**Sec. 8.1** of the supplemental material), avoiding extreme circumstances where the background is heavily occluded and the lighting can be barely identified.
>
> ### Evaluations of generated shadows
>
> > How do the authors evaluate the quality of generated shadows?
>
> As suggested by Reviewer Xnyj, we conduct separate evaluations of foreground and background regions, and the results are shown in **Tab. I** in the attached pdf, where **Ours bg.** performs better than **Original bg.** by a large margin, indicating that the generated shadow makes the repainted background region closer to the reference background. As the low FID score tells us, our background network (NetB) effectively adds a realistic perception to the overall repainting results.
>
> ### Scalability of the proposed method
>
> Please refer to **Sec. II** of the global response.
>
> ### Clarifications
>
> In our experiment setting, no ready foreground objects are awaiting to be harmonized with the background image. The foreground objects are generated **from scratch** following the user-specified conditions. Therefore, strictly speaking, our method is not directly comparable with image harmonization methods. Nevertheless, **Fig. II** in the attached pdf gives an impression of how image harmonization methods handle challenging lighting effects.

---

> ### Author Response · Authors · 2023-08-17
> **Thanks for your comment**
>
> Thank you for your reply and for acknowledging our efforts. (By the way, it appears that your reply might have been posted in the wrong thread.) We understand your concerns about the generalization of our method. We have shown the generalization ability to in-the-wild images in the **Fig. 13** of the originally submitted supplemental material and to casually drawn parsing masks in the **Fig. I** of rebuttal pdf. We will carefully discuss dataset limitations and scalability and incorporate more experimental results in the final version of our paper.
>
> We are eager to hear more valuable suggestions for polishing our paper.

---

### Official Review · Reviewer_Xnyj · 2023-07-06

**Soundness:** 2 fair
**Presentation:** 2 fair
**Contribution:** 2 fair
**Rating:** 3
**Confidence:** 4

**Summary:**

This paper proposes a new method called LuminAIRe to address unrealistic lighting effects in image repainting analogous to cut-and-paste object insertion. This method estimates 3D geometry and environment lighting conditions from background images and parsing masks and uses physically-based illumination rendering and attention modules to inject physically-correct lighting information into the image generation process. The result is repainted images with harmonized lighting effects in both foreground and background regions. To facilitate and validate this task, a new dataset called CAR-LUMINAIRE with lighting annotations and appearance variants has been collected.

**Strengths:**

- Unrealistic lighting effects in image repainting is an open task, especially getting the background shadows right. This paper attempts to have a solution to address it
- The use of physically-based illumination rendering and attention modules is interesting
- Results in repainted images with harmonized lighting effects in both foreground and background regions appear plausible to some extent
- CAR-LUMINAIRE dataset with lighting annotations and appearance variants though not realistic as real images but potentially could be used for prototyping experiments

**Weaknesses:**

- The paper does not compare or cite several relevant previous works [1, 2, 3, 4, 5]
- L48-49 "As far as we know, the illumination-awareness in image editing tasks has not been emphasized yet" is untrue. For instance see [2, 3, 4]
- LuminAIRe pipeline has several components similar to and the current work has completely overlooked previous works and their contributions [see 3]
- The current ablation study does not help figure out which components are adding to improved results. A leave-one-out ablation is crucial to highlight the differences. That is an ablation (a) without Hierarchical Labeling (b) without illumination attention and (c) without illumination injection is necessary.
- There should also be an evaluation comparing foreground and background consistency separately to indicate the difference in the effectiveness for both these regions. A comparison with Image Harmonization methods for the foreground objects and perhaps also with [4] would help clearly understand how these methods compare with image harmonization and reshading methods
- Many of the results look unrealistic and we do not how they compare to other works like [3 or 4] that show examples of the real world.
- Missing real-world examples. Cars have complex material properties with paints and glitter in them. Simple parametric model representation of lighting cannot capture those complex lighting effects. Most of the evaluation is shown in synthetic datasets where the object appearance is mostly diffuse without strong specularities or reflections in them. There is no clear evidence in the current paper that the method would translate well to real scenes as well.
- Results demonstrating the change in appearance (both foreground and background) when the environment's maps are rotated by fixing the camera and scene geometry would also help in understanding how good the results are with changes in lighting conditions.

[1] People as Scene Probes. Wang et al. ECCV 2020 \
[2] Repopulating Street Scenes. Wang et al. CVPR 2021 \
[3] GeoSim: Realistic Video Simulation via Geometry-Aware Composition for Self-Driving. Chen et al. CVPR 2021 \
[4] Cut-and-Paste Object Insertion by Enabling Deep Image Prior for Reshading. Bhattad & Forsyth. 3DV 2022 \
[5] CADSim: Robust and Scalable in-the-wild 3D Reconstruction for Controllable Sensor Simulation. Wang et al. CoRL 2022

**Questions:**

The rebuttal must address the following points raised in weakness:

- Clear comparisons with the methods listed in the missing references
- Recommended ablations
- Real-world examples
- Foreground background separate evaluations
- Correcting claimed contributions on "illumination-aware image editing tasks"
- results demonstrating the change in appearance when environment maps are rotated for the same scene and obects


**Limitations:**

- Missing citations, references, and comparison
- Missing comprehensive ablation
- Missing real-world examples
- overstating claimed contributions and undermining several related work contributions

---

> ### Author Rebuttal · Authors · 2023-08-09
>
> Thanks for your very detailed review and suggestions, and here we give responses to the mentioned concerns.  We are looking forward to discussing with you during the author-reviewer discussion period. Due to the length limit, please refer to the global response for detailed reference items.
> ### More citations needed
> We will add citations and discussions of relevant works mentioned [VII-XI].
>
> [VII] and [VIII] leverage **timelapse image sequences** of street scenes for scene decomposition and extraction of pedestrians and cars (2D image), which are further combined with lighting-based object retrieval,  shadow network, and sun position estimation to composite street view images with illumination-harmonized objects inserted.
>
> [IX] and [XI] use **registered video and LiDAR sensor data** as input to create 3D mesh assets of cars and then insert the 3D car models into the scene with geometry constraints and shadow generation in the video clips for autonomous driving data enhancement.
>
> [X] takes the **background image, foreground image, and mask** of the object to be inserted as input. The reshaded image is lighting-harmonized by forcing the consistent image decomposition using deep image priors and reconstruction supervision.
>
> We respectfully believe that it is unfair to compare our method with [VII], [VIII], [IX], and [XI] due to the **inconsistent inputs**. The only method that shares similar input conditions to us is [X].
>
> We find the starter code of the suggested method [X] is released. However, a time-consuming per-image optimization process (~10 min) is needed for each input image. Therefore we are unable to compare with this work given the limited time during the author response period. We will consider a comparison in the final version.
> ### More ablation results
> As suggested, we test two additional ablation variants: **Ours-A** and **Ours-AI**. The results are shown in **Tab. I** and **Fig. II** in the attached pdf. Please note that there is no “**Ours-I**” (and “**Ours-HI**”) variant since the illumination attention module cannot be enabled alone without illumination injection. The qualitative results show that **Ours-A** gives results of inconsistent lighting effects and wrongly loses/adds highlight effects in the first/second rows, while **Ours-AI** has no awareness about environment lighting and therefore wrongly gives a diffuse appearance/hallucinates lighting effects in the first/second rows. The trends of the quantitative scores also confirm the visual perceptions.
>
>  So far, we have tested all possible ablation variants: **Ours**, **Ours-H**, **Ours-A**, **Ours-AI**, **Ours-HA**, and **Ours-HAI**.
> ### Comparison with image harmonization methods
> As stated above, we cannot compare with [X] during rebuttal time. Instead, we compare with two latest image harmonization methods: **DHT+** [I] and **PCT-Net** [II]. Since there are **no foreground objects to be harmonized in the background images** and we care about to what extent image harmonization methods can recover realistic lighting effects, we use the generated results of **Ours-AI** (without illumination injection in foreground generation) as inputs of  **DHT+** [I] and **PCT-Net** [II].
>
> The results are shown in **Tab. I** and **Fig. II** in the attached pdf. There are reasonable leads in the M-score of **DHT+** [I] and **PCT-Net** [II], indicating better integrity of the harmonized results. The **PCT-Net** [II] also improves the perception of realism compared with **Ours-AI**, as indicated by the decrease in the FID score. However, the degraded the R-prcn and SSIM scores indicate the harmonized images may deviate from the user-specified conditions, as also shown in the qualitative results (**Fig. II**), where the harmonized images do not show better lighting effects and may have severe color shifting issues. We will add such comparisons in the final version.
> ### Real-world examples
> Please refer to **Sec. I** of the global response.
> ### Separate evaluations for foreground and background regions
> We conduct separate quantitative evaluations as suggested. When evaluating the foreground/background regions in the full images, we mask the background/foreground regions with zeros. The quantitative results are reported in **Tab. I** in the attached pdf (the R-prcn and M-score metrics are only meaningful for full images and thus not computed), which shows that our method also outperforms baseline methods wrt. foreground generation and the trends are consistent with the **Realistic** preference of the user study results shown in **Tab. 1** of the main paper. Besides, **Ours bg.** also performs better than **Original bg.** used by baseline methods, showing the effectiveness of our background repainting network (NetB).
> ### Correcting the overclaim
> We are sorry about some overclaim and agree that several prior works on image/video editing have considered illumination, so we will carefully revise our contribution claims in the final version.
> ### Illustrations of rotating estimated lighting
> As suggested, we conduct the experiment of rotating estimated lighting while keeping other conditions untouched. The results are shown in **Fig. III** in the attached pdf. “No light” denotes the repainted results with no illumination information (which is not **Ours-AI**, but **Ours** with illumination disabled at test time), and the degrees (*e.g.* $0\degree/180\degree$) mark the clockwise azimuth rotation angles (in the camera coordinates) of the estimated lighting in the first/second row. The corresponding illumination images are shown as insets. Without the illumination information, the generated foreground becomes flat and has no lighting effects, validating the effectiveness of the illumination injection. As the estimated lighting rotates, the illuminations/appearances of foreground regions correctly reflect the changes in the lighting effects while the repainted backgrounds show reasonable shadows and demonstrate visually realistic perceptions.

---

> > ### Comment · Reviewer_Xnyj · 2023-08-19
> > **Response to author rebuttal**
> >
> > Thank you for the rebuttal and for addressing the concerns I highlighted in my initial review.
> >
> > I revisited the supplementary material, focusing on Fig 13. Unfortunately, the scenes mainly come across as unrealistic. The cars, in particular, have a synthetic and diffuse appearance. They lack the intricate details and complexities we expect from real cars. While I recognize and appreciate your efforts, the appearance in Fig 13 doesn't fully address the concerns about realistic representations of objects.
> >
> > Additionally, Fig III from your rebuttal doesn't effectively illustrate the subtle nuances associated with changing lighting conditions. The variations in lighting and their subsequent effects on the scene are nuanced. This makes it difficult to draw clear conclusions from the figure.
> >
> > The paper's premise is indeed interesting. However, the depth of experimental analysis seems to be lacking. A thorough literature review, and a detailed comparison with related works, particularly methods like cut-and-paste reshading, is crucial. This will help in solidifying the novelty and effectiveness of your approach. I'd like to stress the importance of evaluations that mirror real-world scenarios. I understand that the cut-and-paste reshading method might be time-consuming, taking up to 10 minutes per scene. However, its comparison remains essential. Evaluating your proposed method, even on a smaller test set against the cut-and-paste method, would offer valuable insights. Such an evaluation can shed light on the strengths and potential improvements of your method. It can also position it as a promising approach for the cut-paste insertion task. I'd also suggest a more in-depth qualitative and quantitative evaluation when lighting conditions are rotated, building upon your analysis in Figure III of the rebuttal.
> >
> > Considering the points mentioned, I believe the paper isn't ready for acceptance in its current state. A major revision would be beneficial. I recommend you address these issues thoroughly and think about resubmitting to the next suitable venue.
> >
> > I hope you'll find this feedback constructive.

---

> > > ### Author Response · Authors · 2023-08-19
> > > **Thanks for your comment**
> > >
> > > Thanks for your reply and recognizing our efforts to address your concerns. We will discuss and distinguish the mentioned works [VII-XI] and add experimental results of separated evaluations, more ablations, lighting rotation, and comparisons with relevant single image editing works [I, II, X] as suggested in the final version of our paper. Below we respond to your new comments.
> > >
> > > We are aware that our generated cars in **Fig. 13** of the originally submitted supplemental material and in our Car-LuminAIRe dataset lack some details compared with real cars. These details may be crucial in dedicated car simulation tasks [IX, XI]; however, they are not in our main focus and contribution to **the lighting-realistic generation and perception with user-controlled semantics**, and collecting these details for learning-based training is also far beyond the feasibility of data collection.
> > >
> > > We agree that the cut-and-paste reshading method [X] is a related work to be discussed. However, we must again address the critical difference which makes the comparison essentially unfair (as well as comparisons with image harmonization methods): our method adopts **a lighting-realistic generation** for the repainted foreground region, where **the foreground object is generated faithfully obeying the semantics provided by the parsing mask and attributes**, while the above-mentioned reshading/insertion/harmonization methods [I, II, VII, VIII, X] adopt **a cut-and-paste process** for the foreground content and handle the lighting effects afterward separately, which prevents the users from editing/controlling the object content and requires **a ready-to-use foreground object image from somewhere else** (probably more intricate details and complexities). To sum up, these methods can not handle the lighting-realistic conditional image repainting task proposed in our paper.
> > >
> > > Nevertheless, we have tested several image cases with the cut-and-paste reshading method [X], adopting the same setting used in **Fig. II**, and the comparison on a larger test set is running. In preliminary results, the reshading results tend to be smooth (no highlight effects), the color in the background may be incorrectly baked into the foreground region, and the dark and bright shading changes may be unharmonized with the background lighting. This is likely due to the complex or noisy background images preventing reasonable image intrinsic decomposition, the deep image priors used being less able to represent high-frequency outdoor illumination, and the learned shape priors unable to generalize well to the cars in our dataset. Since no extra images are allowed in comments per the policy, we will add comparison results in the final version as a reference.
> > >
> > > Although we have shown results in **Fig. III** of the rebuttal pdf for validating our pipeline design (where the background content is fixed, the shadings on the car and the shadows on the ground change according to the rotations of the given lighting, showing our method indeed **follows the extracted lighting condition to give lighting-realistic generation**, and removing lighting gives a piecewise-flat repainting image, showing our method indeed **inject the illumination into the generation process**), from the perspective of an image-level editing task, there is no need for rotating the lighting only, since the inconsistency of the background and the rotated lighting would not only confuse the network but also damage the lighting-realistic perception of humans, as **Fig. III** shows. Like the relevant single-image-based works [I, II, X], our method is not designed for the lighting rotation application where a lighting-unharmonized image is the desired output. Nevertheless, we will provide more evaluation results of this setting in the final version for further validation.

---

### Official Review · Reviewer_Ex3g · 2023-07-07

**Soundness:** 3 good
**Presentation:** 4 excellent
**Contribution:** 4 excellent
**Rating:** 7
**Confidence:** 4

**Summary:**


This paper tackles the task of illumination-aware conditional image repainting. Given an input image and a set of conditions, the proposed method aims to inpaint / re-generate a certain region based on the input conditions. This can be used to achieve functionalities such as object insertion and image composition. Compared to prior works, this paper is with the goal of injecting physics-based illumination information into the image generation process.

In a high-level, instead of formulating this task as a simple image-to-image translation in 2D image space, this work aims to introduce explicit physics-based rendering in 3D into a 2D neural renderer. This can be achieved by incorporating physics-based rendering buffers. To enable training and evaluation of the method, the authors also curate a photorealistic synthetic dataset with material and lighting conditions.

The results of the proposed method is qualitative visualized and quantitatively evaluated. A user study is included to compare the photorealism of the edited results. The proposed method can significantly outperform baselines wrt lighting effects.


**Strengths:**

In general I find this paper with a sufficient amount of workload and technically solid.

Originality:

- The task definition is well motivated. The analysis on why we need 3D information in conditional generation is generally informative and convincing.
- The proposed method is sensible and novel. Despite a complicated pipeline, it presents a smart approach to inject physics-based rendering process into a 2D neural renderer.

Quality:

- The qualitative and quantitative results outperform baselines and achieves SOTA.


Clarity:

- This paper is well written and easy to follow.
- The descriptions on method details in main paper and supp are thorough.


Significance:

- This paper proposes a carefully designed approach for illumination-aware image generation, which has not been extensively explored in recent generative models.
- The proposed dataset can be beneficial for future research works.



**Weaknesses:**

In general I do not find critical concerns of this paper but have some questions to further elicit insights:

- The proposed lighting representation is a slightly modified version of prior works. How does the parametric light representation (in Eq.9) compare to prior sky models [22, 23, 32, 63]?
- The model is trained on synthetic data, which can be a concern when the ultimate goal is to apply on real-world imagery. How well does it work on real-world images, and how to measure the domain gap?
- What is the core advantage of generative repainting compared to fully physics-based lighting estimation methods such as SOLID-Net?

The motivation of conditional image repainting is still a relatively small scope. The authors could consider including discussion of these works in related works. For explicit lighting estimation, a line of work estimates 3D lighting volume:
- Wang et al. Neural Light Field Estimation for Street Scenes with Differentiable Virtual Object Insertion
- Li et al. Spatiotemporally Consistent HDR Indoor Lighting Estimation

In a similar spirit to this paper, many works in relighting and neural rendering also combine neural modules with PBR. For example,
- Philip et al. Multi-view Relighting using a Geometry-Aware Network
- Pandey et al. Total Relighting: Learning to Relight Portraits for Background Replacement



**Questions:**

Please see weaknesses section above.

**Limitations:**

The limitations and failure cases are discussed in paper and supp.

---

> ### Author Rebuttal · Authors · 2023-08-09
>
> Thanks for your very detailed review and suggestions, and here we give responses to the mentioned concerns.  We are looking forward to discussing with you during the author-reviewer discussion period.
>
> ### Comparison of different lighting representations
>
> The parameters in **Eq. 9** share many common variables with prior works since these variables are bound with the same physical meanings ($l_\text{sun}$ for sun direction and $c_\text{sun}$ for sun mean color). However, the underlying modeling is differently designed with reasons.
>
> As a common point, the HW sky model [22, 23], LM sky model [32, 63], and our model all adopt the representations that separate the contribution of sun light and sky light in the environment lighting. The HW sky model is derived by modeling light from the sun going through the Earth's atmosphere, which is spectrum-related and somewhat complicated by taking atmospheric scattering into consideration. The LM sky model is derived by directly fitting the environment map of the sky dome (captured on the ground, that is to say, at the bottom of the atmosphere). The sun light part of the LM model is simplified as a double exponential fallout, while the sky light part is modeled using the analytic Preetham sky model [VI].
>
> In our task, we need to represent environment lighting from a complete panoramic view ($360\degree\times180\degree$) and the sky might be occluded. However, the LM and HW models are used to model only the sky dome ($360\degree\times90\degree$) and therefore are not suitable. Since the ambient light may also be from the grounding, where the color variations are much more significant than the sky, we resort to spherical harmonic (SH) for low-frequency ambient light. For representing sun light, we have tried and found the double exponential falloff a little overkill for our lighting data and task requirement. Therefore, we use a simpler spherical Gaussian (SG) modeling for the high-frequency sun light (which is further approximated as directional lighting). The choice of our lighting representation allows us to render the illumination of the object surface with the Blinn-Phong model on the fly efficiently without conducting numerical integral on the surface hemisphere (**Eq. 15** and **Eq. 16** of the supplemental materials) and shallower gradient chain for possible end-to-end training in future work.
>
> [VI]  A Practical Analytic Model for Daylight. Preetham *et al*. SIGGRAPH 1999.
>
> ### Real-world evaluations
>
> > How well does it work on real-world images, and how to measure the domain gap?
>
> We have shown a set of real-world examples in the supplementary material of the original submission, please refer to **Sec. I** of the global response.
>
> For domain gap measurement, the current evaluation protocol cannot be directly used on real-world data due to the lack of ground truth labels, one possibility is to collect a small real-world test set at an affordable cost and then perform the same quantitative evaluation as on the synthetic dataset.
>
> ### Comparison with SOLID-Net
>
> SOLID-Net is a physics-based lighting estimation method that combines the idea of intrinsic image decomposition and differentiable rendering. Their final output is spatially-varying lighting represented as panoramic environment maps.
>
> The repainting results of our methods can be achieved by SOLID-Net using virtual object insertion (VOI). However, even with known lighting, for a realistic VOI, a 3D model along with a properly set shadow catcher is needed, as done in the data collection pipeline of our work. Whereas for realistic repainting, our method only takes a parsing mask along with user-given attributes as input, which is more convenient and has flexibility allowing editing using different attributes.
>
> To sum up, our generative repainting pipeline relieves the demand for 3D models and expert skills in the application of virtual object insertion and gives the users more freedom to control generated content while keeping consistently realistic lighting effects.
>
> ### More citations needed
>
> As suggested, we will add more citations and discussions of works on 3D volumetric lighting estimation and neural PBR relighting.
>
> As for the mentioned papers on  3D volumetric lighting estimation, both papers use the additional depth maps as input, which help the learning of 3D volumetric lighting representation. Wang *et al*.'s paper also utilizes a learned sky dome lighting for long-distance global lighting in outdoor scenes and uses differentiable rendering and adversarial learning techniques to facilitate better lighting estimation for VOI. Li *et al*.'s paper utilizes the spatiotemporal consistent constraint in the video clips by using an RNN to blend the lighting volume estimation results from individual video frames.
>
> As for the mentioned papers on neural PBR relighting, Philip *et al*.'s paper uses a 3D proxy of the outdoor scene reconstructed by multi-view stereo to compute shadow masks and RGB shadow images at the source and target lighting conditions, then obtains refined shadow masks by the refining network, which are used to relighting new scene appearances in a neural rendering module along with illumination buffers from the 3D proxy as other inputs. Pandey *et al*.'s paper aims at relighting portraits given the target environment lighting by conducting intrinsic decomposition to get normal and albedo predictions and then using the given HDR environment map to render a diffuse light map and a set of specular light maps (which has a similar idea as our illumination candidate images, and we'll add a clear reference in the final version), at last, a shading network decides how to combine specular light maps and generates self-shadowing and specularities effects in the relit result.

---

> > ### Comment · Reviewer_Ex3g · 2023-08-16
> >
> > The authors rebuttal responded to my questions and concerns.
> > After reading rebuttal and other reviews, I do agree with other reviewers on limitations of this work. I would encourage the authors to revise accordingly, such as to confront several lines of related work when introducing the task/context, modestly adjust / lower some corresponding claim, and discuss the limitation on the variety of objects in the dataset and scalability. The revision will not weaken the contribution of the paper, but provide more accurate position of this work in the literature.
> >
> > In general I do not find existing concerns critical and could be fixed by revision. I would remain my current rating.

---

> > > ### Author Response · Authors · 2023-08-16
> > > **Thanks for your comment**
> > >
> > > We appreciate your acknowledgment of our efforts to address your concerns. We agree that current concerns can be fixed in the final version of our paper. As suggested, we will carefully revise our paper to confront related work comprehensively, adjust claims modestly, discuss dataset limitations and scalability, and incorporate more experimental results. These changes will enhance our paper's accuracy within the literature without weakening its contribution.
> > >
> > > Your support and recognition of our contribution are greatly encouraging to us.

---

### Official Review · Reviewer_U8wa · 2023-07-07

**Soundness:** 3 good
**Presentation:** 4 excellent
**Contribution:** 3 good
**Rating:** 5
**Confidence:** 4

**Summary:**

The paper proposed a learning-based method for conditional image inpainting. The method
takes an image with a masked region and the conditioned attributes as input, and
synthesizes a new image by filling the masked region. Previous works for this task
usually fail to generate images with realistic shading effects such as specularity
and shadows. The paper tackles this problem by involving geometry and lighting cues  in the inpainting process. The paper estimates lighting information from the input
image and renders shading images of the inpainting content using the estimated lighting,
normal and a set of predefined roughness values. Conditioned on the shading images
and the attributes, additional networks are trained to inpaint the foreground
and background to generate photorealistic shading effects. The paper shows results
on inpainting cars onto outdoor images and demonstrates better results than previous
methods in terms of both FID score and user preference.


**Strengths:**

1. By explicitly predicting lighting, normal and materials, and rendering the shading
images, the proposed method can generate inpainting results that have realistic shading
effects such as shadows and specularity, and outperforms basline methods.

2. The paper performs a thorough ablation study to validate the effectiveness of
different design chocies.


**Weaknesses:**

1. The paper only demonstrates results on a very limited scenario where cars
are inpainted onto outdoor images. The designed lighting representation is also
tailored for outdoor scenes. It's not clear whether the proposed method is scalable
to other cases such as indoor images and more diverse kinds of inpainting content.

2. In terms of shadows, the paper only shows cast shadows on a flat surface. How does
it perform when the cast shadows are on other objects such as walls? In the meantime,
the method can only generate shadows cast by the inpainted content, and cannot
produce shadows on the inpainted content cast by the background objects. For example,
in Figure 13 Row 7 of the supplemental material, there are no shadows on the inpainted car.

3. In the results shown in the paper, all input masks follow a perfect car silhouette.
I am wondering whether this is a requirement of the method. What if a rough mask that
may extend to regions outside of the car is given? What would the normal/shading prediction
look like?

4. The paper should make it clear that it is following previous works on the lighting
representation and add clear references to them. For the illumination image, such a
technique has also been used in previous works such as:
* Deferred Neural Lighting: Free-viewpoint Relighting from Unstructured Photographs
* Total Relighting: Learning to Relight Portraits for Background Replacement

In addition, the paper is also related to inverse rendering tasks in computer graphics,
and it would be good to add discussions on them:
* Inverse Rendering for Complex Indoor Scenes: Shape, Spatially-Varying Lighting and SVBRDF from a Single Image

5. What does the normal mean in the text prompt?

Overall, I think the proposed method is technically sound and the results are convincingly
better than the baseline methods. I refrain from giving a higher rating due to concerns
on the scalability of the method considering that it only shows results on car inpainting
and requires a lot of manually annotated 3D data which may not be easily available for other
subjects such as animals and other general objects.



**Questions:**

See weakness.

**Limitations:**

The limitations look good to me.

---

> ### Author Rebuttal · Authors · 2023-08-09
>
> Thanks for your very detailed review and suggestions, and here we give responses to the mentioned concerns.  We are looking forward to discussing with you during the author-reviewer discussion period.
>
> ### Imperfect parsing masks as input
>
> >  What if a rough mask that may extend to regions outside of the car is given? What would the normal/shading prediction look like?
>
> We conduct an experiment of using disturbed parsing masks (the silhouette extends outside the car, and the internal structure is coarsened with random noise) as input, with the results shown in **Fig. I** in the attached pdf.
>
> In each case, the first row shows results for the accurate parsing mask, while the second row is for the disturbed mask. The other input conditions are kept the same. The results show that the normal and shading predictions for disturbed masks are still reasonable, as well as the final repainting results, which show the robustness of our proposed method to imperfect parsing masks.
>
>  Our repainting formulation follows a pixel-wise correspondence between the semantics in the parsing masks and the repainting results. For example, in the right case, the door handles in the repainted image shrink following the disturbed mask. Therefore, better parsing masks generally lead to better repainting results. On the other hand, the repainting results may be negatively affected given severe mask degradation.
>
> ### Concerns on shadow generation
>
> > The paper only shows cast shadows on a flat surface.
>
> As stated in **Fig. 2** in the main paper and **Sec. 8.1** in the supplementary material of the original submission, the cars are inserted as foreground objects onto a flat “insertable” ground (*e.g.*, roads, grass, and dirt). The shadows of the inserted cars are rendered using a flat shadow catcher fitting the ground. More realistic shadows can be rendered by setting more sophisticated shadow catchers fitting all geometry in the background images,  which is infeasible for a large amount of data with great diversities. Therefore, our shadow net (NetS) only learns to cast shadows on a flat surface with imperfectly rendered shadow labels. Nevertheless, a flat shadow should be reasonably good for open outdoor scenes.
>
> > The method can only generate shadows cast by the inpainted content, and cannot produce shadows on the inpainted content cast by the background objects.
>
> Unlike the cast shadow on the ground, the shadow on the object cast by another object is a highly challenging non-local secondary lighting effect and is very difficult for networks to directly predict by purely operating in the image space. For general outdoor scenes without additional priors, a full 3D reconstruction of the whole scene is needed to recover this type of non-local shadows [V], which usually need multiple view inputs. We will clearly mention such limitations about shadows in the final version.
>
> [V] Neural Fields meet Explicit Geometric Representations for Inverse Rendering of Urban Scenes. Wang *et al*. CVPR 2023.
>
> ### Scalability of the proposed method
>
> Please refer to **Sec. II** of the global response.
>
> ### More citations needed
>
> > The paper should make it clear that it is following previous works on the lighting representation and add clear references to them.
>
> As correctly pointed out, our idea of illumination candidate images is indeed similar to what has been proposed in the two mentioned papers. Thanks for suggesting these relevant works, and we will add clear references to those papers as suggested.
>
> > The paper is also related to inverse rendering tasks in computer graphics, and it would be good to add discussions on them.
>
> We will add more citations and discussions of inverse rendering works as suggested.
>
> As for the specific paper mentioned, Li *et al*.'s paper proposes a single-view method of lighting and BRDF estimation utilizing the learning-based priors on indoor scenes (which might capture scene properties such as Manhattan rooms, local smoothness and global sparsity of the materials, and spatially-consistency of local lighting) from a synthetic rendered dataset (which also require lots of computation, especially for the densely-labeled indoor spatially-varying lighting annotations).
>
> They use 12 SG lobes to represent indoor local lighting near the object surface, which generally works fine for indoor scenes. However, for outdoor scenes with extreme sun intensity, optimizing the ground truth SG parameters may be numerically unstable. Nevertheless, finding a proper lighting representation that works fine in both indoor and outdoor scenes with high efficiency to use is still an open problem.
>
> The commonly used framework of “decomposition-->rendering-->reconstruction loss back-propagation” is more challenging in general outdoor scenes at a single-view setting as the possible priors mentioned may not stand anymore, and it becomes less reliable for the monocular depth and normal estimation, which prohibits precise reconstruction of target image via rendering. In our image repainting setting, we have no accurate “target image” for minimizing reconstruction loss. Therefore, their method cannot directly apply to our task. In our proposed method, we only conduct a forward rendering process where the normal is estimated from the semantics of object categories (cars in our work specifically), and the need for depth is circumvented by the shadow network (NetS).
>
> Due to the incomplete 3D observation of the scene, Li *et al*.'s paper does not handle secondary lighting effects in forward rendering, which is also the limitation of our proposed method.
>
> ### Clarifications
>
> > What does the normal mean in the text prompt?
>
> Sorry for the ambiguity in wording. The “normal” in the text prompt is not related to the concept of “surface normal.” It is closer to the concept of “default values” (for instance, the middle darkness of the glass between light and dark).

---

### Author Rebuttal · Authors · 2023-08-09

We thank the reviewers for their valuable and constructive feedback on our paper, and we are looking forward to a more comprehensive discussion during the author-reviewer discussion period.

We are glad and encouraged to see that the reviewers’ comments that our paper is “well-written and easy to follow” (**Reviewer Ex3g**) and “performs a thorough ablation study” (**Reviewer U8wa**) with “excellent” presentation (**Reviewers U8wa, Ex3g, and Kvqo**) and “a sufficient amount of workload” (**Reviewer Ex3g**); the idea is “interesting” (**Reviewers Xnyj, oBS5, and Kvqo**) and “sound” (**Reviewer Kvqo**); the proposed dataset is “more physically plausible than previous datasets” (**Reviewer Kvqo**), “beneficial for future research work” (**Reviewer Ex3g**) and “useful for illumination community” (**Reviewer oBS5**); and the proposed method is “technically sound” (**Reviewers U8wa and Ex3g**) and “well validated by the experiments and ablation studies” (**Reviewer Kvqo**).

We are also aware that the reviewers have raised many concerns about our paper. Here we post responses for overlapped concerns in global response and individual concerns in dedicated responses for each reviewer, respectively.

In the **attached pdf file**, we provide more experimental results as suggested by the reviewers:

1. Repainting results with rough masks (in **Fig. I**, as suggested by reviewer U8wa).
2. More ablation variants (in **Tab. I** and **Fig. II**, as suggested by reviewer Xnyj).
3. Separated evaluations for foreground and background regions (in **Tab. I**, as suggested by reviewer Xnyj).
4. Comparisons with image harmonization methods (in **Tab. I** and **Fig. II**, as suggested by reviewer Xnyj).
5. Illustrations of appearances as the lighting changes (in **Fig. III**, as suggested by reviewer Xnyj).

In the following text, we address the common issues the reviewers raised:

### I. Real-world examples

We have provided results of real-world examples in **Fig. 13** of the originally submitted supplemental material, where the background images are collected from other datasets [III, IV] and never seen in the whole training process. For other input conditions used in the in-the-wild test, the attributes are randomly generated, and the parsing masks are manually generated for each test image so that the repainted cars are in a suitable place with a proper size. The parsing masks are not hand-drawn but rendered from the 3D annotated car models. To further investigate how the masks influence our results, we provide the results of using rough masks in **Fig. I** of the attached pdf to show how casually-drawn masks would work for our method.

### II. Scalability of the proposed method

In our paper, we only demonstrate the results of car repainting on outdoor scenes so far. This is not limited by our proposed method but by the feasibility of data collection.

The core of our proposed method is injecting extracted lighting information as the illumination images, which would not be affected by changing scenes or object types. On the other hand, our method firmly relies on the learned relationships between semantics and object properties (such as shape, materials, and colors). Therefore, it cannot directly repaint objects with unseen semantics (for example, the semantic of “hairs” for persons does not correspond to any semantic of cars). If enough data can be collected for other object categories, the proposed method should work well. Despite that, we cannot rush a dataset with a qualified amount of data and semantic labeling for other object categories, given the limited period during the author's response.

Since indoor and outdoor scenes have drastically different lightings, prior works often use different representations for indoor [15] and outdoor [21] scenes. Therefore, our proposed method trained on outdoor scenes basically cannot be directly used for inference on indoor scenes (there are only a few works on lighting estimation that work fine on both indoor scenes and outdoor scenes **simultaneously**, and all-scene lighting representation and estimation is still an open problem). For indoor scenes, our current lighting representation could work by only using the SH lighting part and fix $z_\text{vis}=0$. Other representations tailored for indoor scenes (such as SVSG) could be better and also remain compatible with our proposed method.

### References

**In the pdf attachment:**

[I] Transformer for Image Harmonization and Beyond. Guo *et al*. TPAMI 2022.

[II] PCT-Net: Full Resolution Image Harmonization Using Pixel-Wise Color Transformations. Guerreiro *et al*. CVPR 2023.

**In the response texts:**

[III] Scalability in Perception for Autonomous Driving: Waymo Open Dataset. Sun *et al*. CVPR 2020.

[IV] UASOL, A Large-Scale High-Resolution Outdoor Stereo Dataset. Bauer *et al*. Scientific Data 2019.

[V] Neural Fields meet Explicit Geometric Representations for Inverse Rendering of Urban Scenes. Wang *et al*. CVPR 2023.

[VI]  A Practical Analytic Model for Daylight. Preetham *et al*. SIGGRAPH 1999.

[VII] People as Scene Probes. Wang *et al*. ECCV 2020.

[VIII] Repopulating Street Scenes. Wang *et al*. CVPR 2021.

[IX] GeoSim: Realistic Video Simulation via Geometry-Aware Composition for Self-Driving. Chen *et al*. CVPR 2021.

[X] Cut-and-Paste Object Insertion by Enabling Deep Image Prior for Reshading. Bhattad & Forsyth. 3DV 2022.

[XI] CADSim: Robust and Scalable in-the-wild 3D Reconstruction for Controllable Sensor Simulation. Wang *et al*. CoRL 2022.

---

### Decision · Program_Chairs · 2023-09-21

**Decision:**

Accept (poster)

**Comment:**

Reviewers and the AC read the rebuttal and took that into consideration for their final recommendation. Reviewers find this work original and potentially impactful. However, the manuscript needs a clearer discussion of the limitations to provide more accurate position of this work in the literature; the addition of missing citations; and an explicit discussion of similarities and differences to cut-and-paste reshading as well as image harmonization.